# Learning Recursive Multi-Scale Representations for Irregular Multivariate Time Series Forecasting

**Boyuan Li, Zhen Liu, Yicheng Luo, Qianli Ma**[*]
School of Computer Science and Engineering, South China University of Technology
`{csboyuanli,cszhenliu,csluoyicheng2001}@mail.scut.edu.cn`
`qianlima@scut.edu.cn`

## Abstract

Irregular Multivariate Time Series (IMTS) are characterized by uneven intervals between consecutive timestamps, which carry sampling pattern information valuable and informative for learning temporal and variable dependencies. In addition, IMTS often exhibit diverse dependencies across multiple time scales. However, many existing multi-scale IMTS methods use resampling to obtain the coarse series, which can alter the original timestamps and disrupt the sampling pattern information. To address the challenge, we propose ReIMTS, a **Re**cursive multi-scale modeling approach for **I**rregular **M**ultivariate **T**ime **S**eries forecasting. Instead of resampling, ReIMTS keeps timestamps unchanged and recursively splits each sample into subsamples with progressively shorter time periods. Based on the original sampling timestamps in these long-to-short subsamples, an irregularity-aware representation fusion mechanism is proposed to capture global-to-local dependencies for accurate forecasting. Extensive experiments demonstrate an average performance improvement of 27.1% in the forecasting task across different models and real-world datasets. Our code is available at https://github.com/Ladbaby/PyOmniTS.

## 1 Introduction

Multivariate Time Series (MTS) are commonly seen in real-world applications such as healthcare, weather, and biomechanics (Zhang et al., 2023b; Shukla & Marlin, 2021b). While extensive research efforts have been devoted to MTS forecasting task (Nie et al., 2023; Zhang & Yan, 2023; Yu et al., 2024), these methods often assume the input to be regularly sampled and fully observed. In reality, varying sampling rates or schedules can be applied to different series, giving rise to Irregular Multivariate Time Series (IMTS). IMTS forecasting for informed decision-making is challenging due to irregular time intervals within each variable and unaligned observations across variables, where an increasing number of studies have paid attention to (Yalavarthi et al., 2024; Zhang et al., 2024; Mercatali et al., 2024).

Under different temporal resolutions, IMTS can exhibit different patterns reflected in temporal and variable dependencies, similar to hourly, monthly, and yearly patterns in regularly sampled time series. For example, in the healthcare dataset PhysioNet'12 (Silva et al., 2012), IMTS samples contain biomarker (variable) readings for ICU patients during their first 48 hours after admission. In these samples, a 6-hour window corresponds to the common clinical monitoring period (Seymour Christopher W. et al., 2017), while a 24-hour window reflects daily cycles of patients, both of which are useful for assessing disease fluctuations (Klerman et al., 2022; Luo et al., 2025).

Although IMTS can have varying dependencies at different scales under scenarios like healthcare and weather (Menne et al., 2016), capturing them into multi-scale representations while maintaining the original sampling patterns remains challenging. On the one hand, multi-scale methods for MTS typically assume inputs to be regularly sampled and fully observed (Shabani et al., 2023; Wang et al., 2024a; Chen et al., 2024), which are not well-suited for IMTS. On the other hand, multi-scale methods for IMTS are still underexplored (Zhang et al., 2023a; Luo et al., 2025; Marisca et al., 2024).

---

[*]Corresponding author.

These multi-scale IMTS methods often involve resampling to obtain a coarse-grain series, which balances the number of observations across different variables but may disrupt the original sampling pattern. As depicted in Figure 1, the upper sample is from the healthcare dataset PhysioNet'12, while the lower one is downsampled using the same way as (Zhang et al., 2023a). Variable Bilirubin in the original sample shows relatively dense observations in the first 12 hours and sparse observations in the subsequent 36 hours, indicating careful monitoring of disease progression at the beginning of ICU admission (Lakshman et al., 2025; Holford, 2019; Morrill et al., 2020). After downsampling to the coarse series, the dense-to-sparse sampling pattern of variable Bilirubin is disrupted, affecting subsequent dependency learning. Additionally, the disruption of urgent to mild clinical monitoring information is disrupted, where information obtained through more frequent monitoring could be beneficial for early clinical decision-making (Miller et al., 2007).

To preserve essential sampling pattern information during multi-scale dependency learning in the above scenarios, we propose ReIMTS, a recursive multi-scale approach for IMTS forecasting. At each scale level, ReIMTS splits an IMTS sample into smaller subsamples with shorter time periods, while maintaining the original sampling timestamps for all observations and thus preserving the original sampling pattern. By recursively splitting the sample in a top-down manner, input IMTS are viewed from a global to local perspective. Each backbone captures dependencies within a specific scale level, and learned latent representations are transferred from higher scale levels to lower ones. Leveraging global-to-local multi-scale representations learned from preserved sampling patterns, ReIMTS employs an irregularity-aware fusion mechanism to capture semantics across different scales, thereby providing accurate forecasting results. Moreover, ReIMTS is compatible with most existing IMTS models due to its flexible architectural design, boosting their forecasting performance in a plug-and-play way.

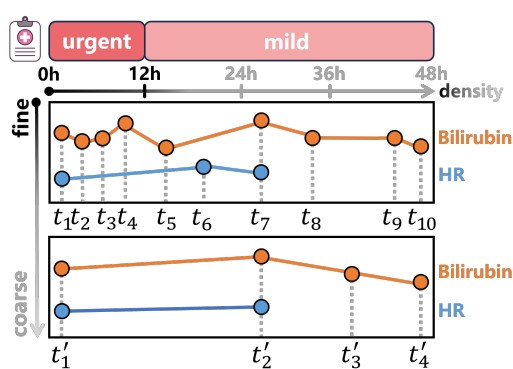

Figure 1: Existing multi-scale methods for IMTS resample the coarse series to balance differences in sampling densities across different variables. In the original sample from the healthcare dataset PhysioNet'12, liver function marker Bilirubin and heart rate (HR) exhibit a dense-to-sparse sampling pattern reflecting urgent to mild clinical monitoring, which is disrupted in the coarse series.

Our main contributions are as follows:

- We introduce recursive splitting based on time periods for IMTS samples to preserve the original sampling patterns across all scale levels, and leverage IMTS backbones to capture dependencies in different time periods as multi-scale representations.

- We propose ReIMTS, a recursive multi-scale method for IMTS forecasting. Using irregularity-aware representation fusion, it recursively captures global-to-local dependencies and provides accurate predictions.

- Extensive experiments including twenty-six baseline methods and five IMTS datasets on IMTS forecasting are conducted. Tested on six IMTS backbones, ReIMTS consistently boosts their forecasting performance in all settings while maintaining good efficiency.

## 2 RELATED WORK

### 2.1 IRREGULAR MULTIVARIATE TIME SERIES FORECASTING

In recent years, an increasing number of studies have paid attention to IMTS forecasting (Yalavarthi et al., 2024; Zhang et al., 2024; Mercatali et al., 2024). From a model architecture perspective, methods for IMTS modeling can be broadly categorized into RNN-based (Che et al., 2018; Shukla & Marlin, 2019), ODE-based (Rubanova et al., 2019; Biloš et al., 2021; Mercatali et al., 2024), GNN-based (Yalavarthi et al., 2024; Zhang et al., 2022; Luo et al., 2025), Set-based (Horn et al., 2020), Diffusion-based (Tashiro et al., 2021), and Transformer-based (Zhang et al., 2023a). While a variety

of model architectures have been employed in IMTS modeling, most of them follow the encoder-decoder structure. Therefore, inputs for their decoders can include temporal representations (Che et al., 2018; Shukla & Marlin, 2019; Rubanova et al., 2019; Biloš et al., 2021; Mercatali et al., 2024), variable representations (Luo et al., 2024; Zhang et al., 2024; Luo et al., 2025; Marisca et al., 2024), observation representations (Yalavarthi et al., 2024; Zhang et al., 2022; Horn et al., 2020), or combinations thereof.

## 2.2 MULTI-SCALE MODELING FOR TIME SERIES

Existing methods for regularly sampled time series have widely adopted multi-scale information during modeling for accurate predictions. From a sampling pattern perspective, Pyraformer (Liu et al., 2022), NHITS (Challu et al., 2023), Scaleformer (Shabani et al., 2023), and TimeMixer (Wang et al., 2024a) use embedding merging or resampling that disrupts original observed timestamps to obtain different scales, potentially missing out on sampling pattern information. Pathformer (Chen et al., 2024) and MOIRAI (Woo et al., 2024) segment time series based on the number of observations rather than time periods, which cannot preserve the sampling rate information. TAMS-RNNs (Chen et al., 2021) was designed based on RNNs for regularly sampled time series, thus not well adapted for IMTS. Multi-scale modeling in IMTS methods is relatively underexplored. Warpformer (Zhang et al., 2023a), Hi-Patch (Luo et al., 2025), and HD-TTS (Marisca et al., 2024) address irregularities within IMTS, but they also employ resampling to get different scales and still cannot preserve the original sampling patterns.

## 3 PROBLEM DEFINITION

With a total of $T$ timestamps and $V$ variables, an IMTS sample can be denoted as a set containing $Y$ observation tuples $\mathbf{S} := \{(t_i, z_i, v_i)|i = 1, ..., Y\}$, where $t_i \in \{0, ..., T\}$, $z_i \in \mathbb{R}$, and $v_i \in \{1, ..., V\}$ represents the timestamp, observed value, and variable indicator respectively. For the IMTS forecasting task, the set of forecast queries $\mathbf{Q} := \{q_j|j = 1, ..., Y_Q\}$ containing $Y_Q$ observations is derived by combining $(t_j, v_j)$ of the $j$-th observation tuple within the forecast window. We aim to learn a forecasting model $\mathcal{F}(\cdot)$, such that given an input IMTS sample $\mathbf{S}$ and a forecast query $\mathbf{Q}$ as input, it accurately predicts the corresponding observed values $\mathbf{Z}$:

$$\mathcal{F}(\mathbf{S}, \mathbf{Q}) \to \mathbf{Z}. \tag{1}$$

## 4 METHODOLOGY

The overview of our proposed method, ReIMTS, is illustrated in Figure 2. We first explain how to learn representations recursively at different scale levels in Section 4.1. Subsequently, we detail the irregularity-aware representation fusion in Section 4.2. Training loss design is described in Section 4.3. Finally, we discuss the differences between our method and existing approaches in Section 4.4. Details on the operations are available at Algorithm 1. Further discussion of how backbones address irregularities and forecast-related queries can be found in the Appendix C.

## 4.1 RECURSIVE LEARNING ACROSS SCALES IN IMTS

In this section, we explain the methods for obtaining representations at each scale level, and how we ensure that the shape of global representations at the current scale matches the shape of local representations in the subsequent scale. It should be noted that in the following discussion, 'global' and 'local' are used to describe relative scales between upper and lower levels, rather than considering all $N$ levels. During data preprocessing, we align the raw multivariate time series within each sample $\mathbf{S}$ based on timestamps to obtain the zero-padded sample $\mathbf{S}^1 \in \mathbb{R}^{L^1 \times V}$ at scale level 1, where $L^1$ is the maximum number of observations in a univariate time series. Also, a corresponding mask $\mathbf{M}^1 \in \{0, 1\}^{L^1 \times V}$ is created, indicating actual observations with 1s and zero-padded values with 0s. As shown in the left panel of Figure 2, for each scale level $n \in \{1, \ldots, N\}$ among the total $N$ levels, an IMTS sample $\mathbf{S}^1$ is recursively partitioned by time periods to generate a series of subsamples. At scale level $n$, we denote the length of time period as $T^n$, the number of subsamples as $P^n = \frac{T^1}{T^n}$, and the maximum number of observations in a univariate time series after splitting and zero padding

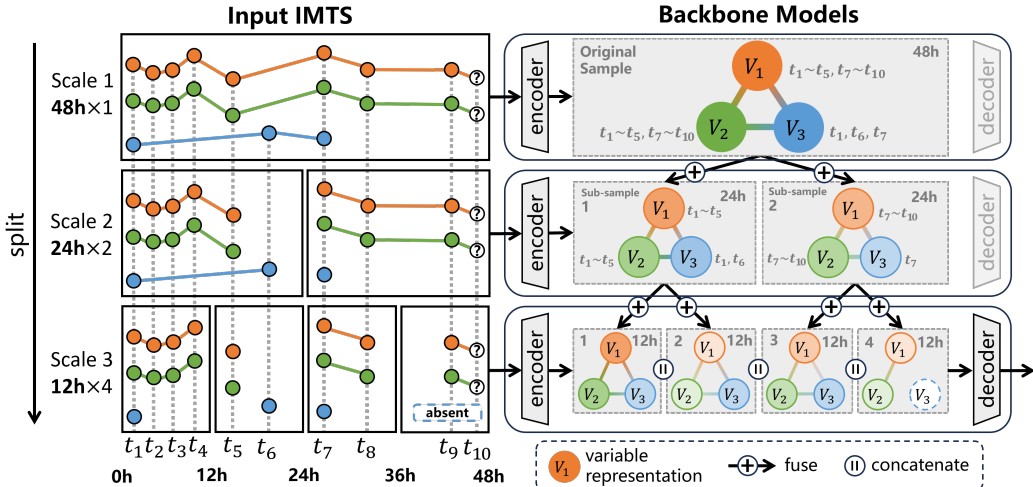

Figure 2: The architecture of ReIMTS with three scale levels. For the original IMTS sample on the top left, ReIMTS recursively splits it into subsamples with shorter time periods at each scale level. ReIMTS is compatible with most IMTS models, and we use graph neural networks as backbones here to illustrate multi-scale variable representation learning. Local representations from lower scale levels are fused with global ones from upper scale levels. The decoder in the lowest scale concatenates representations and decodes them into forecast predictions.

as $L^n$. For the $k$-th subsample $\mathbf{s}(\mathbf{t}_k^n)$ with $k \in \{1, ..., P^n\}$, its time period consists of the timestamps $\mathbf{t}_k^n = \{t | T^n(k-1) < t \leq T^n k\}$. Therefore, the set of all subsamples at scale level $n$ can be written as:

$$\mathbf{S}^n := \{\mathbf{s}(\mathbf{t}_k^n)\}_{k=1}^{P^n}, \tag{2}$$

where $\mathbf{S}^n \in \mathbb{R}^{P^n \times L^n \times V}$. The mask is splitted in the same way to obtain $\mathbf{M}^n$. It should be noted that $L^n$ and $T^n$ are distinct: whereas $T^n$ incorporates real-world time units such as minutes, hours, or years, $L^n$ merely denotes the number of observations without any time units. It should also be noted that the split position is based on time periods rather than an equal number of observations, where the term 'observation' includes both actual observed values and zero-padded values used for alignment. As noted in previous studies (Chowdhury et al., 2023; Zhang et al., 2024), splitting IMTS samples based on the number of observations can result in subsamples that correspond to different time lengths in reality. This can affect the learning of varying sampling density information in the original data. Therefore, we use a time period splitting approach to preserve the sampling information across all scale levels. The specific time periods chosen for each dataset are described in Appendix B.2.

At scale level $n$, the IMTS backbone $\mathcal{F}^n(\cdot)$ uses its encoder $\mathcal{F}_{\text{enc}}^n(\cdot)$ to obtain latent representations $\mathbf{E}^n$ of input subsamples $\mathbf{S}^n$:

$$\mathbf{E}^n = \mathcal{F}_{\text{enc}}^n(\mathbf{S}^n). \tag{3}$$

The definition of the encoder can vary across different IMTS backbones, with our implementation details provided in Appendix C. For most existing IMTS models, latent representations fall into three categories $\mathbf{E}^n \in \{\mathbf{E}_{\text{time}}^n, \mathbf{E}_{\text{var}}^n, \mathbf{E}_{\text{obs}}^n\}$: temporal representations $\mathbf{E}_{\text{time}}^n \in \mathbb{R}^{P^n \times L^n \times D}$, variable representations $\mathbf{E}_{\text{var}}^n \in \mathbb{R}^{P^n \times V \times D}$, and observation representations $\mathbf{E}_{\text{obs}}^n \in \mathbb{R}^{P^n \times L^n \times V \times D}$. Here, $D$ denotes the hidden dimension.

During subsequent processing, $\mathbf{E}^n$ is first transformed into $\mathbf{G}^n$ through irregularity-aware representation fusion, and then reshaped into $\mathbf{H}^n$ to match the shape of $\mathbf{E}^{n+1}$ at the next scale. To compute $\mathbf{G}^n$ from $\mathbf{E}^n$, we incorporate global representations from the upper scale when $n > 1$, as detailed in Section 4.2. For $n = 1$, we simply set $\mathbf{G}^n = \mathbf{E}^n$. Accordingly, $\mathbf{G}^n$ also consists of the three types $\mathbf{G}^n \in \{\mathbf{G}_{\text{time}}^n, \mathbf{G}_{\text{var}}^n, \mathbf{G}_{\text{obs}}^n\}$, each retaining the same shape as its corresponding counterpart in $\mathbf{E}^n$. To further obtain $\mathbf{H}^n$ from $\mathbf{G}^n$, we follow the implementation described in Appendix A. Specifically, we split along the temporal dimension for temporal or observation representations, yielding $\mathbf{H}_{\text{time}}^n$ or $\mathbf{H}_{\text{obs}}^n$, and apply duplication for variable representations to obtain $\mathbf{H}_{\text{var}}^n$. The resulting output

$\mathbf{H}^n \in \{\mathbf{H}^n_{\text{time}}, \mathbf{H}^n_{\text{var}}, \mathbf{H}^n_{\text{obs}}\}$ at scale $n$ is then provided to scale $n+1$ as its global representation. In the following section, we discuss how global-to-local representations are recursively fused across scales.

## 4.2 IRREGULARITY-AWARE REPRESENTATION FUSION

In this section, we introduce the irregularity-aware recursive fusion of global-to-local representations. At the lower scale level $n+1$, we want to evaluate the importance of the global representation $\mathbf{H}^n$ from upper scale level $n$, while accounting for the inherent irregularity in the original IMTS. Therefore, a lightweight scoring layer is utilized to assign weights $\alpha$ to the global representation $\mathbf{H}^n$. Moreover, the binary mask $\mathbf{M}^{n+1}$ is also used to indicate irregularity.

$$\mathbf{H}^n_{\text{IMTS}} = \begin{cases} \mathbf{H}^n \cdot \mathbf{M}^{n+1}, & \text{when } \mathbf{H}^n = \mathbf{H}^n_{\text{time}} \text{ or } \mathbf{H}^n_{\text{obs}} \\ \mathbf{H}^n, & \text{when } \mathbf{H}^n = \mathbf{H}^n_{\text{var}} \end{cases}, \tag{4}$$

$$\alpha = \text{ReLU}(\text{FF}(\mathbf{H}^n_{\text{IMTS}})), \tag{5}$$

where ReLU is the non-linear activation function and FF is a feed-forward layer. It should be noted that the irregularity information for variable representations $\mathbf{H}^n_{\text{var}}$ has been encoded by the encoders of IMTS backbones, while padding values are still present in observation representations $\mathbf{H}^n_{\text{time}}$ and $\mathbf{H}^n_{\text{obs}}$. Therefore, we use the mask $\mathbf{M}^{n+1}$ to distinguish between observations and padding values specifically in temporal and observation representations. The score $\alpha$ is then used to fuse the local representation $\mathbf{E}^{n+1}$ and global one:

$$\mathbf{G}^{n+1} = \mathbf{E}^{n+1} + \alpha \mathbf{H}^n_{\text{IMTS}}. \tag{6}$$

## 4.3 TRAINING OF ReIMTS

In this section, we introduce the process for obtaining forecast values and training ReIMTS. At the lowest scale level $N$, the decoder of IMTS backbone $\mathcal{F}_{\text{dec}}$ takes the concatenated representation as input and predicts the forecast values $\hat{\mathbf{Z}}$:

$$\hat{\mathbf{Z}} = \mathcal{F}_{\text{dec}}(\text{Concat}(\{\mathbf{G}^n\}_{n=1}^N)), \tag{7}$$

where the definition of decoder is the subsequent network modules after the encoder of IMTS backbone, and Concat denotes the concatenation of representations from the same sample. For a detailed explanation of the backbone decoder's structure, please refer to Appendix C. The model is trained by minimizing the Mean Squared Error (MSE) loss between the predicted values $\hat{\mathbf{Z}}$ for forecast queries and their corresponding ground truth $\mathbf{Z}$:

$$\mathcal{L} = \frac{1}{Y_Q} \sum_{j=1}^{Y_Q} (\hat{z}_j - z_j)^2. \tag{8}$$

It should be noted that only forecast queries are used in loss calculations, which is implemented by multiplying prediction $\hat{\mathbf{Z}}$ and binary masks $\mathbf{M}_Q \in \{0, 1\}^{L_Q \times V}$ corresponding to the forecast horizon $L_Q$.

## 4.4 DELINEATING FROM EXISTING METHODS

We discuss the differences and similarities between ReIMTS and existing approaches in this section, and depicted in Figure 3. Patch-based methods, such as tPatchGNN (Zhang et al., 2024) and PrimeNet (Chowdhury et al., 2023), can be viewed as variants of ReIMTS with only one scale level. Although they split samples based on time periods, tPatchGNN and PrimeNet are limited to a single time period length during training. In contrast, ReIMTS can choose varying lengths at different scale levels, allowing for the exploration of richer multi-scale dependencies. Moreover, ReIMTS is a plug-and-play method that seamlessly works with most encoder-decoder IMTS backbones. It should be noted that tPatchGNN and PrimeNet learn global dependencies through inter-patch learning, while ReIMTS learns from its upper scale levels. Compared to multi-scale methods for regularly sampled time series such as Pathformer (Chen et al., 2024), MOIRAI (Woo et al., 2024), and Scaleformer (Shabani et al., 2023), the assumption of regular sampling and division by the number

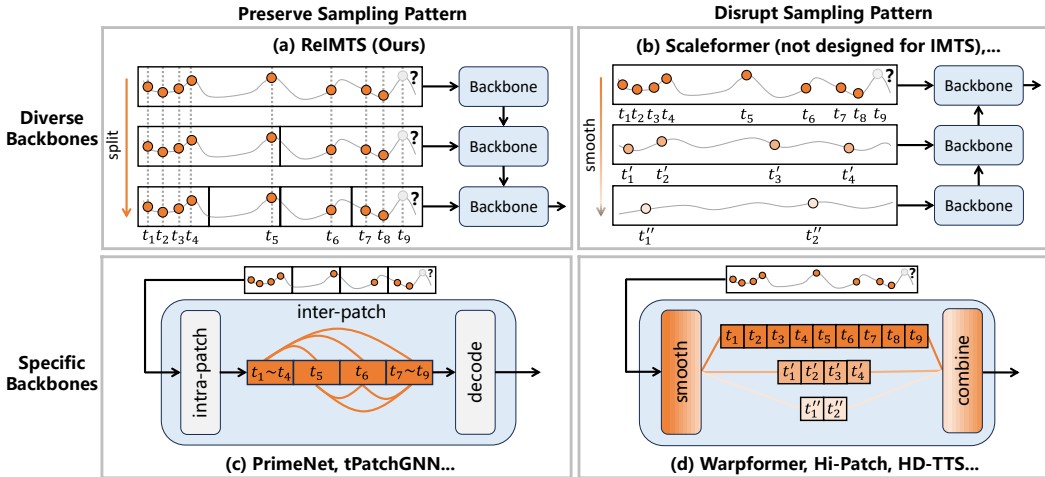

Figure 3: Comparison of our method and existing approaches. (a) ReIMTS preserves the original sampling pattern while remaining compatible with most IMTS backbones. (b) Sample-space resampling methods. (c) Patch-based methods for IMTS. (d) Representation-space resampling methods.

Table 1: Summary of five irregular time series datasets.

| Description | MIMIC-III | MIMIC-IV | PhysioNet'12 | Human Activity | USHCN |
|---|---|---|---|---|---|
| Max length | 96 | 971 | 47 | 131 | 337 |
| # Variable | 96 | 100 | 36 | 12 | 5 |
| # Sample | 21,250 | 17,874 | 11,981 | 1,359 | 1,114 |
| Avg # obs. | 144.6 | 304.8 | 308.6 | 362.2 | 313.5 |
| Avg # obs. (padding) | 9,216.0 | 92,000.0 | 1,692.0 | 1,573.2 | 1,685.0 |

of observations makes them not well-suited, as described in Section 4.1. Furthermore, Pathformer and MOIRAI require slow transformer-based layers, whereas ReIMTS can utilize lightweight IMTS backbones such as GraFITi (Yalavarthi et al., 2024) or mTAN (Shukla & Marlin, 2021a). Scaleformer uses resampling to obtain multi-scale inputs, which can disrupt the sampling pattern information, as discussed in Section 1.

## 5 EXPERIMENTS

### 5.1 EXPERIMENTAL SETUP

#### 5.1.1 DATASETS

Five widely studied irregular multivariate time series datasets, covering healthcare, biomechanics, and climate, are used in the experiments. Their statistics are summarized in Table 1. MIMIC-III (Johnson et al., 2016) is a clinical database collected from ICU patients during the initial 48 hours of admission, which is rounded for 30 minutes. MIMIC-IV (Johnson et al., 2023) is built upon MIMIC-III, which has a higher sampling frequency and data are rounded for 1 minute. PhysioNet'12 (Silva et al., 2012) is also a clinical database collected during the first 48 hours of ICU stay, rounded for 1 hour. Human Activity includes biomechanical data detailing 3D positional variables, which are rounded for 1 millisecond. USHCN (Menne et al., 2016) includes climate data spanning over 150 years, collected from meteorological stations distributed across the United States. Our analysis focuses on a subset of 4 years between 1996 and 2000. For all five datasets, we follow the preprocessing setup provided in the publicly available code pipeline PyOmniTS (Li et al., 2025) v2.0.0, which originated from GraFITi and tPatchGNN (Yalavarthi et al., 2024; Zhang et al., 2024). It splits datasets into training, validation, and test sets adhering to ratios of 8:1:1, a common split setting used in previous works (Zhang et al., 2022; Luo et al., 2025).

Table 2: Experimental results for our method (+**ReIMTS**) with respective baselines, evaluated by MSE (mean $\pm$ std) $\times 10^{-1}$ on five irregular multivariate time series datasets. The best results are indicated in **bold**. Average improvements (error reductions) are marked with $\uparrow$.

| Algorithm | | MIMIC-III | MIMIC-IV | PhysioNet'12 | Human Activity | USHCN | Vs Ours |
|---|---|---|---|---|---|---|---|
| PrimeNet | original | $9.04 \pm 0.00$ | $6.25 \pm 0.00$ | $7.93 \pm 0.00$ | $26.84 \pm 0.02$ | $4.57 \pm 0.00$ | |
| | +**ReIMTS** | $\mathbf{4.76 \pm 0.19}$ | $\mathbf{3.58 \pm 0.03}$ | $\mathbf{3.01 \pm 0.03}$ | $\mathbf{0.82 \pm 0.02}$ | $\mathbf{1.71 \pm 0.06}$ | $\uparrow\mathbf{62.3\%}$ |
| mTAN | original | $8.51 \pm 0.14$ | $5.09 \pm 0.12$ | $3.75 \pm 0.02$ | $0.89 \pm 0.03$ | $5.65 \pm 0.67$ | |
| | +**ReIMTS** | $\mathbf{6.37 \pm 0.05}$ | $\mathbf{4.04 \pm 0.10}$ | $\mathbf{3.51 \pm 0.02}$ | $\mathbf{0.89 \pm 0.01}$ | $\mathbf{1.70 \pm 0.01}$ | $\uparrow\mathbf{24.3\%}$ |
| TimeCHEAT | original | $4.41 \pm 0.05$ | $2.50 \pm 0.01$ | $3.27 \pm 0.11$ | $0.68 \pm 0.04$ | $1.73 \pm 0.04$ | |
| | +**ReIMTS** | $\mathbf{4.40 \pm 0.03}$ | $\mathbf{2.02 \pm 0.03}$ | $\mathbf{2.90 \pm 0.01}$ | $\mathbf{0.52 \pm 0.01}$ | $\mathbf{1.62 \pm 0.06}$ | $\uparrow\mathbf{12.1\%}$ |
| GRU-D | original | $4.75 \pm 0.04$ | $5.97 \pm 0.22$ | $\mathbf{3.25 \pm 0.00}$ | $1.76 \pm 0.23$ | $2.42 \pm 0.17$ | |
| | +**ReIMTS** | $\mathbf{4.67 \pm 0.05}$ | $\mathbf{3.91 \pm 0.10}$ | $\mathbf{3.25 \pm 0.00}$ | $\mathbf{0.51 \pm 0.01}$ | $1.89 \pm 0.08$ | $\uparrow\mathbf{25.8\%}$ |
| Raindrop | original | $5.13 \pm 0.02$ | $3.41 \pm 0.05$ | $3.27 \pm 0.01$ | $0.89 \pm 0.02$ | $2.04 \pm 0.07$ | |
| | +**ReIMTS** | $\mathbf{5.05 \pm 0.06}$ | $\mathbf{2.95 \pm 0.06}$ | $\mathbf{3.14 \pm 0.01}$ | $\mathbf{0.87 \pm 0.00}$ | $\mathbf{2.01 \pm 0.05}$ | $\uparrow\mathbf{4.5\%}$ |
| GraFITi | original | $4.08 \pm 0.02$ | $2.39 \pm 0.01$ | $2.85 \pm 0.01$ | $0.43 \pm 0.00$ | $1.71 \pm 0.05$ | |
| | +**ReIMTS** | $\mathbf{4.07 \pm 0.02}$ | $\mathbf{1.79 \pm 0.05}$ | $\mathbf{2.83 \pm 0.01}$ | $\mathbf{0.42 \pm 0.00}$ | $\mathbf{1.66 \pm 0.02}$ | $\uparrow\mathbf{6.3\%}$ |

### 5.1.2 BASELINES

We perform the comparisons also using code pipeline PyOmniTS (Li et al., 2025) v2.0.0. Twenty-six baselines are included in the benchmark, covering SOTA methods categorized as (1) Multi-scale methods for IMTS: HD-TTS (Marisca et al., 2024), Hi-Patch (Luo et al., 2025), Warpformer (Zhang et al., 2023a), (2) Other SOTA methods for IMTS: TimeCHEAT (Liu et al., 2025), GNeuralFlow (Mercatali et al., 2024), tPatchGNN (Zhang et al., 2024), GraFITi (Yalavarthi et al., 2024), PrimeNet (Chowdhury et al., 2023), CRU (Schirmer et al., 2022), Raindrop (Zhang et al., 2022), NeuralFlows (Biloš et al., 2021), mTAN (Shukla & Marlin, 2021a), SeFT (Horn et al., 2020), GRU-D (Che et al., 2018) (3) Multi-scale methods for regularly sampled time series: Ada-MSHyper (Shang et al., 2024), MOIRAI (Woo et al., 2024), TimeMixer (Wang et al., 2024a), Pathformer (Chen et al., 2024), Scaleformer (Shabani et al., 2023), (4) Other SOTA methods for regularly sampled time series: Leddam (Yu et al., 2024), PatchTST (Nie et al., 2023), TimesNet (Wu et al., 2023), Crossformer (Zhang & Yan, 2023), Autoformer (Wu et al., 2021). We adapt their publicly available codes into the pipeline for comparisons, where network structures remain unchanged.

### 5.1.3 IMPLEMENTATION DETAILS

We follow the setting of widely acknowledged Time-Series-Library (Wang et al., 2024b) in learning rate adjustments. All experiments run with a maximum of 300 epochs and early stopping patience of 10 epochs. To mitigate randomness, we conduct each experiment with five different random seeds ranging from 2024 to 2028 also following Time-Series-Library, calculating both the mean and standard deviation of the results. MSE is used as the training loss function for models, unless a custom loss function proposed in the original paper is used. When adapting regular time series models for IMTS, masks indicating observed values are included in the MSE calculations during training. The detailed settings for the hyperparameters are provided in Appendix E. Due to our more fine-grained hyperparameter searches for each experimental settings, baselines can perform better than those reported in previous works (Li et al., 2025; Yalavarthi et al., 2024; Zhang et al., 2024). All models are trained on a Linux server with PyTorch version 2.7.0 and two NVIDIA GeForce RTX 3090 GPUs, while the efficiency analysis is conducted on another Linux server with PyTorch version 2.2.2+cu118 and one NVIDIA GeForce RTX 2080Ti GPU.

## 5.2 MAIN RESULTS

Table 2 compares the ReIMTS version of existing IMTS models with respective baselines, and Table 3 shows the models' forecasting performance. Both are evaluated using MSE across five datasets, with the best results highlighted in bold. The visualization of forecasting results can be found in Appendix D. The lookback time periods are 36 hours for MIMIC-III, MIMIC-VI, and PhysioNet'12, 3000 milliseconds for Human Activity, and 3 years for USHCN. Human Activity uses a forecast length of 300 milliseconds, and the rest datasets use the next 3 timestamps as forecast targets, following

Table 3: Experimental results for other state-of-the-art regular and irregular baselines on five irregular multivariate time series datasets evaluated by MSE (mean $\pm$ std) $\times 10^{-1}$. The best and second-best results are indicated in **bold** and underlined, respectively. 'ME' indicates memory error.

|  | Algorithm | MIMIC-III | MIMIC-IV | PhysioNet'12 | Human Activity | USHCN |
|---|---|---|---|---|---|---|
| Regular | MOIRAI | $8.66 \pm 0.00$ | $4.29 \pm 0.00$ | $4.92 \pm 0.00$ | $1.08 \pm 0.00$ | $12.14 \pm 0.00$ |
|  | Ada-MSHyper | $6.16 \pm 0.01$ | $3.89 \pm 0.01$ | $4.06 \pm 0.02$ | $1.48 \pm 0.04$ | $2.36 \pm 0.07$ |
|  | Autoformer | $7.08 \pm 0.08$ | $5.45 \pm 0.16$ | $4.14 \pm 0.04$ | $0.98 \pm 0.07$ | $4.12 \pm 0.29$ |
|  | Scaleformer | $5.50 \pm 0.07$ | $4.55 \pm 0.13$ | $4.02 \pm 0.02$ | $1.01 \pm 0.05$ | $5.18 \pm 0.96$ |
|  | TimesNet | $5.78 \pm 0.01$ | $3.82 \pm 0.01$ | $4.08 \pm 0.01$ | $1.21 \pm 0.03$ | $2.20 \pm 0.09$ |
|  | NHITS | $5.83 \pm 0.01$ | $3.95 \pm 0.01$ | $3.84 \pm 0.01$ | $0.98 \pm 0.01$ | $3.09 \pm 0.19$ |
|  | Pyraformer | $5.69 \pm 0.02$ | $4.08 \pm 0.04$ | $3.82 \pm 0.00$ | $1.17 \pm 0.01$ | $1.90 \pm 0.03$ |
|  | PatchTST | $5.68 \pm 0.01$ | $2.94 \pm 0.01$ | $3.40 \pm 0.01$ | $0.68 \pm 0.00$ | $2.18 \pm 0.07$ |
|  | Leddam | $5.93 \pm 0.00$ | $3.70 \pm 0.02$ | $3.75 \pm 0.03$ | $0.91 \pm 0.01$ | $2.38 \pm 0.10$ |
|  | Pathformer | $5.59 \pm 0.13$ | ME | $3.46 \pm 0.01$ | $0.91 \pm 0.01$ | $5.69 \pm 1.39$ |
|  | Crossformer | $5.37 \pm 0.01$ | $3.00 \pm 0.02$ | $3.39 \pm 0.05$ | $1.41 \pm 0.21$ | $1.87 \pm 0.05$ |
|  | TimeMixer | $5.67 \pm 0.05$ | $3.54 \pm 0.02$ | $3.25 \pm 0.01$ | $0.67 \pm 0.02$ | $2.92 \pm 0.54$ |
| Irregular | SeFT | $9.23 \pm 0.01$ | $6.60 \pm 0.00$ | $7.67 \pm 0.01$ | $13.76 \pm 0.02$ | $4.12 \pm 0.02$ |
|  | NeuralFlows | $7.17 \pm 0.03$ | $4.74 \pm 0.02$ | $4.20 \pm 0.02$ | $1.68 \pm 0.03$ | $3.89 \pm 0.04$ |
|  | CRU | $7.07 \pm 0.03$ | $4.35 \pm 0.02$ | $6.19 \pm 0.01$ | $1.37 \pm 0.04$ | $2.35 \pm 0.04$ |
|  | GNeuralFlow | $6.95 \pm 0.05$ | $5.01 \pm 0.02$ | $3.88 \pm 0.03$ | $1.73 \pm 0.01$ | $2.59 \pm 0.05$ |
|  | tPatchGNN | $5.17 \pm 0.04$ | $2.74 \pm 0.02$ | $3.22 \pm 0.02$ | $0.44 \pm 0.01$ | $2.11 \pm 0.13$ |
|  | Hi-Patch | $4.35 \pm 0.02$ | $2.36 \pm 0.02$ | $3.11 \pm 0.05$ | $0.48 \pm 0.01$ | $2.34 \pm 0.11$ |
|  | Warpformer | $4.09 \pm 0.01$ | $2.42 \pm 0.02$ | $2.88 \pm 0.01$ | $0.54 \pm 0.01$ | $1.77 \pm 0.03$ |
|  | HD-TTS | $4.17 \pm 0.01$ | $2.36 \pm 0.00$ | **$2.83 \pm 0.01$** | $0.50 \pm 0.01$ | $1.66 \pm 0.04$ |
|  | **ReIMTS (Ours)** | **$4.07 \pm 0.02$** | **$1.79 \pm 0.05$** | **$2.83 \pm 0.01$** | **$0.42 \pm 0.00$** | **$1.66 \pm 0.02$** |

the settings in existing works (Biloš et al., 2021; De Brouwer et al., 2019). Results evaluated using MAE are detailed in Appendix B.3, and an analysis of varying forecast horizons can be found in Appendix B.4. As can be seen, our method ReIMTS boosts the performance of six existing IMTS models by an average of 27.1%, including well-known models and SOTA ones. When using GraFITi as the backbone, ReIMTS achieves the overall best performance compared to all baseline models. Compared to existing multi-scale methods for IMTS such as Hi-Patch, Warpformer, and HD-TTS, ReIMTS demonstrates superior extensibility by integrating multi-scale techniques without requiring a dedicated multi-scale module within the backbone. We also observe that older methods, such as mTAN and GRU-D, can also achieve performance improvements and even outperform more recent models. This demonstrates the potential of enhancing classic methods for IMTS forecasting with our approach. PrimeNet demonstrates significant performance improvements after applying ReIMTS, which may suggest that its pretraining-finetuning approach require more input samples to achieve optimal performance, and we further discuss it in Appendix B.4. As for regularly sampled time series models, some of them can surpass relatively old IMTS models, which shows the necessity of comprehensive comparisons.

Table 4: Ablation results of ReIMTS and its four variants on five irregular multivariate time series datasets using GraFITi as the backbone.

| Ablation | MIMIC-III | MIMIC-IV | PhysioNet'12 | Human Activity | USHCN |
|---|---|---|---|---|---|
| ReIMTS | **$4.07 \pm 0.02$** | **$1.79 \pm 0.05$** | **$2.83 \pm 0.01$** | **$0.42 \pm 0.00$** | **$1.66 \pm 0.02$** |
| rp sample | $4.99 \pm 0.05$ | $1.92 \pm 0.04$ | $2.83 \pm 0.01$ | $0.45 \pm 0.01$ | $1.69 \pm 0.03$ |
| rp split | $5.02 \pm 0.04$ | $2.36 \pm 0.03$ | $3.20 \pm 0.01$ | $0.61 \pm 0.02$ | $2.31 \pm 0.09$ |
| rp IARF | $4.20 \pm 0.02$ | $1.84 \pm 0.02$ | $2.79 \pm 0.00$ | $0.47 \pm 0.01$ | $1.89 \pm 0.05$ |
| w/o IARF | $4.77 \pm 0.08$ | $2.07 \pm 0.04$ | $3.06 \pm 0.00$ | $0.54 \pm 0.00$ | $1.69 \pm 0.04$ |

## 5.3 EFFICIENCY ANALYSIS

We compare the efficiency of our method ReIMTS with existing multi-scale methods for IMTS, namely Warpformer (Zhang et al., 2023a), HD-TTS (Marisca et al., 2024), and Hi-Patch (Luo et al., 2025). GraFITi (Yalavarthi et al., 2024) is used as the backbone in ReIMTS, and we also perform a comparison with it. Models are assessed based on their MSE, training time, and GPU memory footprint. The training time for one epoch with a batch size of 32 is recorded, then divided by the number of batches to determine the training time per iteration. Memory footprints only encompass the model's usage instead of representing the entire process. Results on the MIMIC-III dataset are shown in Figure 4, and results from the remaining datasets are plotted in Figure 6 and discussed in Appendix B.5. As can be seen, ReIMTS runs the fastest and uses the least GPU memory compared to existing multi-scale methods for IMTS, while achieving the lowest MSE. It demonstrates ReIMTS's flexibility in using lightweight backbones and scalability in performing the best. Compared with the original backbone GraFITi, ReIMTS controls the overhead of multi-scale learning within an acceptable range, without the GPU memory usage being proportional to the number of scale levels.

## 5.4 ABLATION STUDY

We evaluate the performance of ReIMTS and its four variants across all five datasets. (1) **rp sample** replaces split subsamples with original sample; (2) **rp split** splits subsamples based on the number of observations rather than time periods; (3) **rp IARF** replaces irregularity-aware representation fusion with addition; (4) **w/o IARF** removes irregularity-aware representation fusion; The ablation results are summarized in Table 4. As can be seen, all model designs are necessary. Results from **rp sample** and **rp split** show the necessity of splitting samples, particularly by time periods rather than the number of observations. **rp IARF** and **w/o IARF** demonstrate the effectiveness of irregularity-aware representation fusion, and highlight the necessity of leveraging both global and local representations.

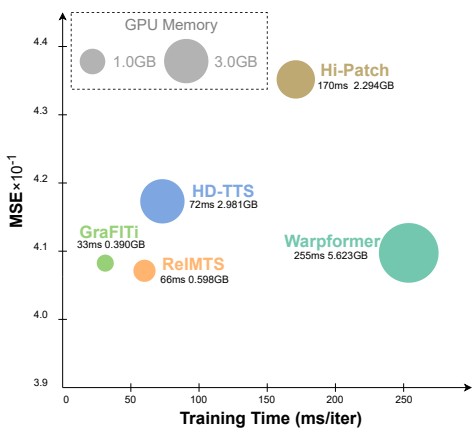

Figure 4: Model efficiency comparison on MIMIC-III, with a 36-hour lookback length, 3 forecast timestamps, 96 variables, and a batch size of 32. ReIMTS uses GraFITi as backbone in the figure, which achieves the best efficiency compared to other multi-scale IMTS methods, including Warpformer, HD-TTS, and Hi-Patch.

## 6 CONCLUSION

This paper introduces a recursive multi-scale method, ReIMTS, to address the IMTS forecasting problem. ReIMTS recursively divides original IMTS samples into subsamples with shorter time periods while maintaining the original sampling patterns. By recursively invoking the backbone to learn representations at each scale level, ReIMTS retrieves global-to-local multi-scale representations based on the preserved sampling patterns. Moreover, ReIMTS leverages an irregularity-aware representation fusion mechanism to adaptively combine global and local representations based on content and sampling patterns, thereby preserving crucial semantics for accurate forecasting. ReIMTS exhibits strong flexibility and improves performance across various existing IMTS backbones, outperforming twenty-six baseline models in our unified code pipeline. Nevertheless, ReIMTS still has more potential in combining with wider range of backbones. Although structurally compatible with most encoder-decoder IMTS models, ODE-based models might need further theoretical explainability when used with our method. Additionally, some diffusion-based models predict noisy latent representations in their backbones, which may not be directly compatible with our method. Possible solutions include predicting clean observations during the denoising process, or using ReIMTS within the denoising backbone. We will investigate these challenges further in future work.

## ACKNOWLEDGEMENTS

We thank the anonymous reviewers for their helpful feedbacks, and all the donors of the original datasets. The work described in this paper was funded by the National Key R&D Program of China (Grant No. 2023YFA1011601), the National Natural Science Foundation of China (Grant Nos. 62272173, 62273109), the Natural Science Foundation of Guangdong Province (Grant Nos. 2024A1515010089, 2022A1515010179), and the Science and Technology Planning Project of Guangdong Province (Grant No. 2023A0505050106).

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

---

**Algorithm 1** ReIMTS: Learning Recursive Multi-Scale Representations

---

**Require:** input IMTS $\mathbf{S}^1 \in \mathbb{R}^{L^1 \times V}$, binary mask $\mathbf{M}^1 \in \{0,1\}^{L^1 \times V}$, scale level $n \in \{1,...,N\}$,
   a series of time periods $\{T^n\}_{n=1}^N$, forecast horizon $L_Q$,
   IMTS backbone decoder $\mathcal{F}_{\text{dec}}$, and a series of encoders in IMTS backbones $\{\mathcal{F}_{\text{enc}}^n\}_{n=1}^N$.
**Ensure:** forecast result $\hat{\mathbf{Z}} \in \mathbb{R}^{L_Q \times V}$
   **function** REIMTS($\mathbf{S}^n, \mathbf{M}^n, n, N, \{T^n\}_{n=1}^N, L_Q, \{\mathcal{F}_{\text{enc}}^n\}_{n=1}^N, \mathcal{F}_{\text{dec}}, \mathbf{H}^{n-1}$)
      $\mathbf{E}^n \leftarrow \mathcal{F}_{\text{enc}}^n(\mathbf{S}^n, L_Q)$                                    ▷ Equation (3) of the paper
      **if** $\mathbf{H}^{n-1} \neq$ null **then**
         $\mathbf{G}^n = \text{Fuse}(\mathbf{E}^n, \mathbf{H}^{n-1}, \mathbf{M}^n)$                    ▷ Equation (4), (5), and (6) of the paper
      **else**
         $\mathbf{G}^n = \mathbf{E}^n$
      **end if**
      **if** $n = N$ **then**
         $\hat{\mathbf{Z}} \leftarrow \mathcal{F}_{\text{dec}}^n(\mathbf{G}^n, L_Q)$                                    ▷ Base case
         **return** $\hat{\mathbf{Z}}$
      **else**
         $\mathbf{S}^{n+1} \leftarrow \text{Split}(\mathbf{S}^n, T^{n+1})$                              ▷ Equation (2) of the paper
         $\mathbf{M}^{n+1} \leftarrow \text{Split}(\mathbf{M}^n, T^{n+1})$
         $\mathbf{H}^n \leftarrow \text{SplitOrDuplicate}(\mathbf{G}^n)$              ▷ Equation (9), (10), and (11) of the paper
         REIMTS($\mathbf{S}^{n+1}, \mathbf{M}^{n+1}, n+1, N, \{T^n\}_{n=1}^N, L_Q, \{\mathcal{F}_{\text{enc}}^n\}_{n=1}^N, \mathcal{F}_{\text{dec}}, \mathbf{H}^n$)  ▷ Recursive call
      **end if**
   **end function**
   **procedure** MAIN
      $\hat{\mathbf{Z}} \leftarrow$ REIMTS($\mathbf{S}^1, \mathbf{M}^1, 1, N, \{T^n\}_{n=1}^N, L_Q, \{\mathcal{F}_{\text{enc}}^n\}_{n=1}^N, \mathcal{F}_{\text{dec}}, \text{null}$)
      **return** $\hat{\mathbf{Z}}$
   **end procedure**

---

## A   RECURSIVE REPRESENTATION SPLITTING OR DUPLICATION

After $\mathbf{E}^n$ is fused with the representation from scale level $n-1$ to obtain $\mathbf{G}^n$ as detailed in Section 4.2, for the temporal representation case $\mathbf{G}_{\text{time}}^n := \{\mathbf{g}_{\text{time}}^n(\mathbf{t}_k^n)\}_{k=1}^{P^n}$ where $\mathbf{G}_{\text{time}}^n \in \mathbb{R}^{P^n \times L^n \times D}$, ReIMTS splits them along the time dimension before passing these global representations to the lower $n+1$ scale level. It should be noted that in this discussion, 'global' and 'local' are used to describe relative scales between upper and lower levels, rather than considering all $N$ levels. The splitting process is similar to that for original IMTS samples, with split positions determined by time periods:

$$\mathbf{H}_{\text{time}}^n := \{\mathbf{g}_{\text{time}}^n(\mathbf{t}_{k'}^{n+1})\}_{k'=1}^{P^{n+1}}, \tag{9}$$

where $\mathbf{H}_{\text{time}}^n \in \mathbb{R}^{P^{n+1} \times L^{n+1} \times D}$. It should be noted that although representations from different levels share the same timestamps, the dependencies they learn differ. Global dependencies $\mathbf{H}_{\text{time}}^n$ are learned from time periods of length $T^n$, while local ones $\mathbf{E}_{\text{time}}^{n+1}$ correspond to shorter time periods of length $T^{n+1}$.

The observation representation case $\mathbf{G}_{\text{obs}}^n := \{\mathbf{g}_{\text{obs}}^n(\mathbf{t}_k^n)\}_{k=1}^{P^n}$, where $\mathbf{G}_{\text{obs}}^n \in \mathbb{R}^{P^n \times L^n \times V \times D}$, is similar to the temporal representation case. We also split them using time periods:

$$\mathbf{H}_{\text{obs}}^n := \{\mathbf{g}_{\text{obs}}^n(\mathbf{t}_{k'}^{n+1})\}_{k'=1}^{P^{n+1}}, \tag{10}$$

where $\mathbf{H}_{\text{obs}}^n \in \mathbb{R}^{P^{n+1} \times L^{n+1} \times V \times D}$. As for the variable representation case $\mathbf{G}_{\text{var}}^n := \{\mathbf{g}_{\text{var}}^n(\mathbf{t}_k^n)\}_{k=1}^{P^n}$ where $\mathbf{G}_{\text{var}}^n \in \mathbb{R}^{P^n \times V \times D}$, we duplicate the number of representations $P^n$ at scale level $n$ for $\left\lceil \frac{P^{n+1}}{P^n} \right\rceil$ times, resulting in the same number of representations $P^{n+1}$ at scale level $n+1$:

$$\mathbf{H}_{\text{var}}^n := \{\mathbf{g}_{\text{var}}^n(\mathbf{t}_{k'}^{n+1})\}_{k'=1}^{P^{n+1}}, \tag{11}$$

where $\mathbf{H}_{\text{var}}^n \in \mathbb{R}^{P^{n+1} \times V \times D}$.

# B  ADDITIONAL EXPERIMENTS

## B.1  EFFECT OF SCALE LEVELS

Table 5: Effect of the number of scale levels on five datasets. A larger number of scale levels performs better on datasets with longer maximum lengths and a sufficient number of samples.

| Scale level | MIMIC-III | MIMIC-IV | PhysioNet'12 | Human Activity | USHCN |
|---|---|---|---|---|---|
| 2 | $4.09 \pm 0.03$ | $\mathbf{1.79 \pm 0.05}$ | $2.86 \pm 0.01$ | $\mathbf{0.42 \pm 0.00}$ | $\mathbf{1.66 \pm 0.02}$ |
| 3 | $\mathbf{4.07 \pm 0.02}$ | $1.95 \pm 0.05$ | $\mathbf{2.83 \pm 0.01}$ | $0.71 \pm 0.00$ | $1.80 \pm 0.03$ |
| 4 | $4.10 \pm 0.02$ | $2.00 \pm 0.02$ | $2.84 \pm 0.01$ | $0.44 \pm 0.00$ | $2.01 \pm 0.02$ |

We assess how the number of scale levels impacts model performance, with results summarized in Table 5. Time periods in different scale levels are 24h, 12h, and 6h for the healthcare datasets MIMIC-III, MIMIC-IV, and PhysioNet'12, 2000ms, 1000ms, and 500ms for Human Activity, and 2 years, 1 year, and 6 months for USHCN. These chosen time periods are based on prior domain knowledge, including cycles in clinical medicine (Lakshman et al., 2025; Holford, 2019; Morrill et al., 2020) and periodic changes in climate (Almazroui et al., 2012). In general, most datasets achieve optimal performance at a scale level of 2, except for PhysioNet'12 and MIMIC-III. Although the USHCN dataset has a relatively long maximum length, it suffers from an insufficient number of samples for training a robust model, as reported in previous work (Yalavarthi et al., 2024).

## B.2  EFFECT OF TIME PERIODS

The impact of varying time periods on model performance is demonstrated in Figure 5 for all datasets. We set the number of scale levels to three on MIMIC-III and PhysioNet'12, and two on the rest, following the result in Appendix B.1. We only change the time period at the second scale level. As can be seen, time periods corresponding to optimal performance across all datasets are half the total time length when the number of scale levels is two. This is expected given that most IMTS datasets are sparse, exhibiting dependencies over long time periods, as discussed in Appendix B.1. Specifically, for the medical datasets MIMIC-III, MIMIC-IV, and PhysioNet'12, a 24-hour time length corresponds to the daily cycles of patients.

## B.3  DIFFERENT METRICS

We present the results measured using MAE in Table 6, which follows the same experimental setup as Table 3. As can be seen, ReIMTS still outperforms all twenty-six baselines across all five datasets. Additionally, it consistently enhances the performance of existing IMTS models under all settings, aligning with the findings from the MSE evaluations discussed in Section 5.2.

## B.4  VARYING FORECAST HORIZONS

We also evaluate performance across different forecast horizons. We follow the same settings as in existing work (Zhang et al., 2024) and keep the lookback length settings the same as in Table 3. For MIMIC-III, MIMIC-IV, and PhysioNet'12, the forecast horizon is set to 12 hours. For Human Activity, the forecast horizon is 1000 milliseconds. For USHCN, the forecast horizon is set to 1 year. The results are summarized in Table 7. It is evident that ReIMTS consistently enhances performance across longer forecast lengths. Regarding other baseline models, we have surprisingly noticed that a few models can outperform others in forecasting longer time horizons, as the empirical results have been thoroughly verified. For example, PrimeNet performs better on longer forecast lengths, suggesting that apart from requiring more input samples, its pretraining-finetuning approach may also require more training targets in forecast horizons to achieve optimal performance. It should be noted that, unlike fully observed MTS, IMTS samples are typically split into lookback and forecast window based on time periods rather than the number of observations. The forecast settings here view 75% of the time period as the lookback window, while the remaining 25% as the forecast window.

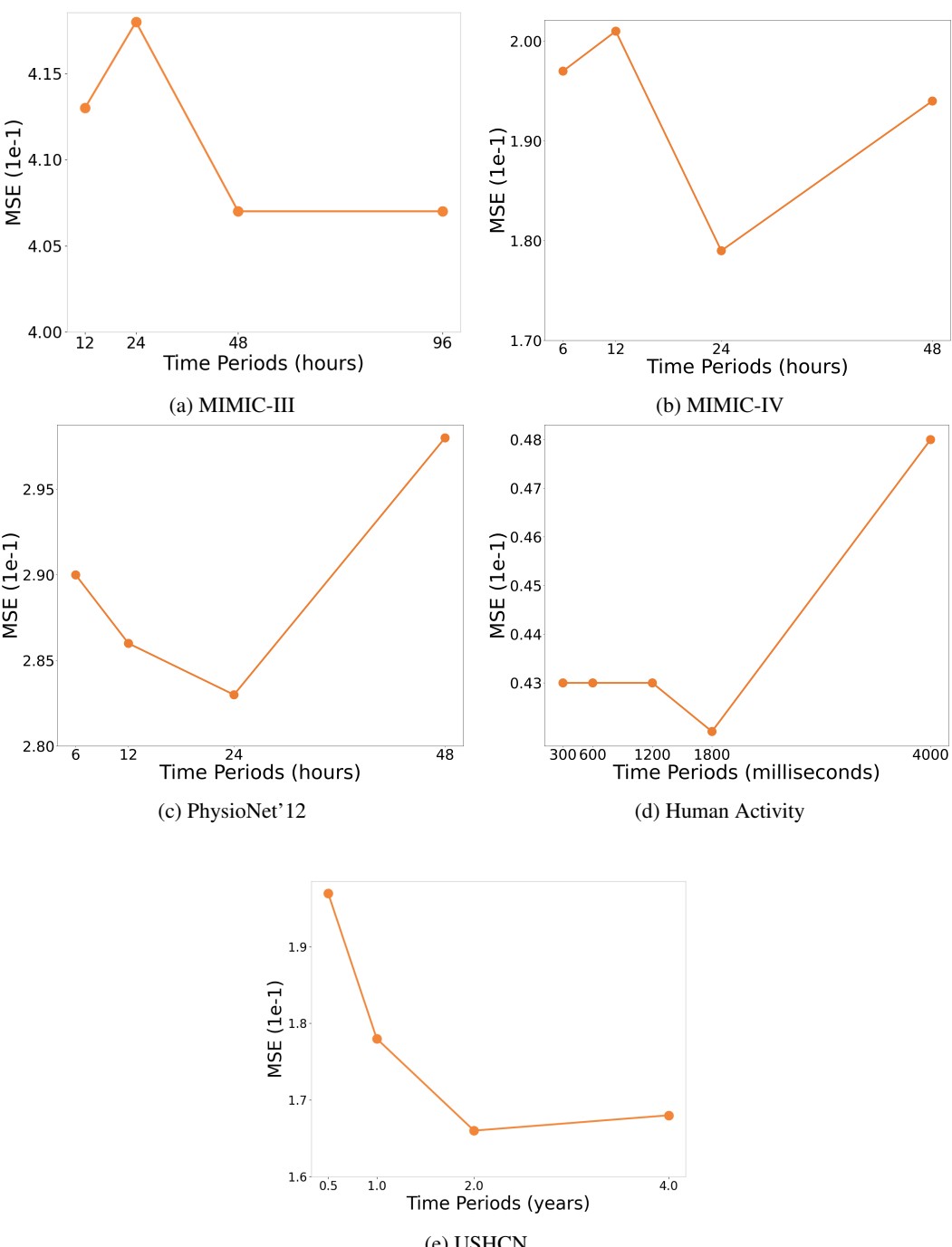

Figure 5: Effect of different time period lengths on PhysioNet'12, Human Activity, and USHCN.

Table 6: Experimental results on five irregular multivariate time series datasets, evaluated using MAE (mean $\pm$ std) $\times 10^{-1}$. The experimental setup is the same as in Table 3. 'ME' indicates Memory Error.

| | Algorithm | MIMIC-III | MIMIC-IV | PhysioNet'12 | Human Activity | USHCN | Vs Ours |
|---|---|---|---|---|---|---|---|
| Regular | MOIRAI | $5.79 \pm 0.00$ | $4.00 \pm 0.00$ | $4.94 \pm 0.00$ | $1.91 \pm 0.00$ | $8.18 \pm 0.00$ | |
| | Ada-MSHyper | $5.22 \pm 0.00$ | $4.28 \pm 0.01$ | $4.50 \pm 0.00$ | $2.61 \pm 0.04$ | $3.25 \pm 0.07$ | |
| | Autoformer | $5.47 \pm 0.03$ | $5.20 \pm 0.00$ | $4.53 \pm 0.03$ | $2.18 \pm 0.06$ | $3.84 \pm 0.12$ | |
| | Scaleformer | $4.93 \pm 0.03$ | $4.64 \pm 0.08$ | $4.45 \pm 0.02$ | $2.25 \pm 0.05$ | $3.37 \pm 0.12$ | |
| | TimesNet | $5.03 \pm 0.01$ | $4.25 \pm 0.01$ | $4.54 \pm 0.01$ | $2.26 \pm 0.02$ | $3.19 \pm 0.04$ | |
| | NHITS | $4.92 \pm 0.01$ | $4.13 \pm 0.01$ | $4.29 \pm 0.01$ | $2.16 \pm 0.01$ | $3.63 \pm 0.06$ | |
| | Pyraformer | $5.08 \pm 0.00$ | $4.53 \pm 0.00$ | $4.28 \pm 0.01$ | $2.28 \pm 0.00$ | $3.01 \pm 0.01$ | |
| | PatchTST | $4.54 \pm 0.01$ | $3.25 \pm 0.01$ | $3.90 \pm 0.00$ | $1.70 \pm 0.01$ | $3.07 \pm 0.06$ | |
| | Leddam | $4.83 \pm 0.03$ | $3.98 \pm 0.02$ | $4.28 \pm 0.02$ | $2.00 \pm 0.01$ | $3.06 \pm 0.05$ | |
| | Pathformer | $4.75 \pm 0.02$ | ME | $4.01 \pm 0.01$ | $1.89 \pm 0.02$ | $4.99 \pm 0.44$ | |
| | Crossformer | $4.74 \pm 0.00$ | $3.59 \pm 0.01$ | $3.95 \pm 0.05$ | $2.50 \pm 0.17$ | $2.74 \pm 0.06$ | |
| | TimeMixer | $4.75 \pm 0.04$ | $3.88 \pm 0.02$ | $3.80 \pm 0.01$ | $1.66 \pm 0.03$ | $3.45 \pm 0.31$ | |
| Irregular | PrimeNet | $6.59 \pm 0.00$ | $5.73 \pm 0.00$ | $6.79 \pm 0.00$ | $13.27 \pm 0.01$ | $4.79 \pm 0.00$ | |
| | SeFT | $6.62 \pm 0.00$ | $5.88 \pm 0.01$ | $6.68 \pm 0.01$ | $9.75 \pm 0.01$ | $4.41 \pm 0.04$ | |
| | mTAN | $6.19 \pm 0.06$ | $5.01 \pm 0.05$ | $4.30 \pm 0.00$ | $2.18 \pm 0.03$ | $5.18 \pm 0.14$ | |
| | NeuralFlows | $5.49 \pm 0.01$ | $4.79 \pm 0.01$ | $4.60 \pm 0.01$ | $3.09 \pm 0.04$ | $3.39 \pm 0.04$ | |
| | CRU | $5.37 \pm 0.01$ | $4.56 \pm 0.01$ | $5.82 \pm 0.01$ | $2.57 \pm 0.04$ | $3.19 \pm 0.06$ | |
| | TimeCHEAT | $4.12 \pm 0.03$ | $2.97 \pm 0.01$ | $3.70 \pm 0.00$ | $1.70 \pm 0.06$ | $2.70 \pm 0.03$ | |
| | GNeuralFlow | $5.35 \pm 0.03$ | $4.90 \pm 0.00$ | $4.38 \pm 0.03$ | $3.15 \pm 0.02$ | $3.38 \pm 0.04$ | |
| | GRU-D | $4.53 \pm 0.02$ | $5.47 \pm 0.15$ | $3.91 \pm 0.01$ | $3.15 \pm 0.20$ | $2.90 \pm 0.03$ | |
| | Raindrop | $4.44 \pm 0.01$ | $3.88 \pm 0.03$ | $3.94 \pm 0.01$ | $2.09 \pm 0.03$ | $3.04 \pm 0.12$ | |
| | tPatchGNN | $4.18 \pm 0.00$ | $3.10 \pm 0.02$ | $3.83 \pm 0.02$ | $1.24 \pm 0.00$ | $2.87 \pm 0.10$ | |
| | Hi-Patch | $4.08 \pm 0.01$ | $2.88 \pm 0.01$ | $3.71 \pm 0.04$ | $1.25 \pm 0.02$ | $2.86 \pm 0.07$ | |
| | Warpformer | $3.90 \pm 0.01$ | $2.97 \pm 0.01$ | $3.54 \pm 0.01$ | $1.30 \pm 0.01$ | $2.74 \pm 0.04$ | |
| | HD-TTS | $3.94 \pm 0.00$ | $2.94 \pm 0.01$ | $\underline{3.49 \pm 0.01}$ | $1.39 \pm 0.04$ | $2.74 \pm 0.05$ | |
| | GraFITi | $\underline{3.74 \pm 0.01}$ | $3.07 \pm 0.02$ | $\overline{3.52 \pm 0.01}$ | $\underline{1.20 \pm 0.00}$ | $2.75 \pm 0.13$ | |
| ReIMTS | +mTAN | $5.17 \pm 0.03$ | $4.23 \pm 0.05$ | $3.96 \pm 0.01$ | $2.11 \pm 0.01$ | $\underline{2.67 \pm 0.02}$ | ↑**18.3%** |
| | +GRU-D | $4.29 \pm 0.00$ | $4.27 \pm 0.07$ | $3.83 \pm 0.00$ | $1.43 \pm 0.01$ | $\overline{2.99 \pm 0.07}$ | ↑**16.2%** |
| | +Raindrop | $4.40 \pm 0.02$ | $3.64 \pm 0.05$ | $3.76 \pm 0.01$ | $1.97 \pm 0.01$ | $3.04 \pm 0.08$ | ↑**3.5%** |
| | +PrimeNet | $4.34 \pm 0.00$ | $4.04 \pm 0.01$ | $3.64 \pm 0.02$ | $2.02 \pm 0.00$ | $2.71 \pm 0.03$ | ↑**47.6%** |
| | +TimeCHEAT | $4.10 \pm 0.04$ | $2.56 \pm 0.06$ | $3.51 \pm 0.01$ | $1.47 \pm 0.01$ | $2.70 \pm 0.08$ | ↑**6.6%** |
| | +GraFITi | $\mathbf{3.69 \pm 0.01}$ | $\mathbf{2.36 \pm 0.01}$ | $\mathbf{3.48 \pm 0.01}$ | $\mathbf{1.15 \pm 0.01}$ | $\mathbf{2.66 \pm 0.05}$ | ↑**6.6%** |

Table 7: Experimental results of varying forecast horizons on five irregular multivariate time series datasets evaluated using MSE (mean $\pm$ std) $\times 10^{-1}$, with the lookback length following Table 3 and forecast horizons set to the rest length of the whole series, which are 12 hours for MIMIC-III, MIMIC-IV, and PhysioNet'12, 1000 milliseconds for Human Activity, and 1 year for USHCN. 'ME' indicates memory error.

| | Algorithm | MIMIC-III | MIMIC-IV | PhysioNet'12 | Human Activity | USHCN | Vs Ours |
|---|---|---|---|---|---|---|---|
| **Regular** | MOIRAI | $9.68 \pm 0.00$ | $4.75 \pm 0.00$ | $5.36 \pm 0.00$ | $1.02 \pm 0.00$ | $8.85 \pm 0.00$ | |
| | Ada-MSHyper | $6.73 \pm 0.05$ | $4.30 \pm 0.03$ | $4.46 \pm 0.02$ | $1.60 \pm 0.01$ | $4.30 \pm 0.03$ | |
| | Scaleformer | $7.36 \pm 0.08$ | $4.84 \pm 0.11$ | $4.34 \pm 0.03$ | $1.54 \pm 0.04$ | $4.97 \pm 0.06$ | |
| | NHITS | $7.43 \pm 0.02$ | $5.16 \pm 0.20$ | $4.60 \pm 0.01$ | $1.10 \pm 0.01$ | $4.52 \pm 0.09$ | |
| | Pyraformer | $6.49 \pm 0.02$ | $4.71 \pm 0.00$ | $4.30 \pm 0.01$ | $1.23 \pm 0.01$ | $3.99 \pm 0.03$ | |
| | PatchTST | $5.90 \pm 0.01$ | $3.36 \pm 0.00$ | $3.91 \pm 0.01$ | $0.87 \pm 0.01$ | $4.28 \pm 0.04$ | |
| | Pathformer | $6.08 \pm 0.10$ | ME | $3.88 \pm 0.01$ | $0.91 \pm 0.03$ | $6.54 \pm 0.08$ | |
| | TimeMixer | $5.43 \pm 0.02$ | $3.71 \pm 0.05$ | $3.76 \pm 0.01$ | $0.69 \pm 0.01$ | $5.50 \pm 0.64$ | |
| **Irregular** | PrimeNet | $8.85 \pm 0.00$ | $5.90 \pm 0.00$ | $7.89 \pm 0.00$ | $10.94 \pm 0.00$ | $7.32 \pm 0.00$ | |
| | mTAN | $9.35 \pm 0.32$ | $4.95 \pm 0.12$ | $4.16 \pm 0.02$ | $1.01 \pm 0.02$ | $5.63 \pm 0.49$ | |
| | TimeCHEAT | $4.89 \pm 0.03$ | $3.02 \pm 0.01$ | $3.61 \pm 0.30$ | $0.74 \pm 0.03$ | $3.96 \pm 0.03$ | |
| | GNeuralFlow | $7.43 \pm 0.07$ | $4.83 \pm 0.00$ | $4.39 \pm 0.02$ | $1.89 \pm 0.08$ | $4.86 \pm 0.09$ | |
| | GRU-D | $5.53 \pm 0.05$ | $4.45 \pm 0.00$ | $3.81 \pm 0.01$ | $1.83 \pm 0.23$ | $6.13 \pm 0.30$ | |
| | Raindrop | $5.82 \pm 0.02$ | $5.58 \pm 0.25$ | $3.81 \pm 0.01$ | $0.97 \pm 0.04$ | $4.49 \pm 0.19$ | |
| | tPatchGNN | $5.90 \pm 0.05$ | $2.88 \pm 0.00$ | $3.81 \pm 0.01$ | $0.60 \pm 0.01$ | $6.19 \pm 0.16$ | |
| | Hi-Patch | $5.03 \pm 0.02$ | $2.97 \pm 0.02$ | $3.81 \pm 0.00$ | $\underline{0.59 \pm 0.00}$ | $4.43 \pm 0.18$ | |
| | Warpformer | $4.83 \pm 0.02$ | $2.99 \pm 0.00$ | $3.62 \pm 0.01$ | $\underline{0.61 \pm 0.01}$ | $\underline{3.90 \pm 0.02}$ | |
| | HD-TTS | $5.62 \pm 0.07$ | $2.84 \pm 0.02$ | $3.78 \pm 0.00$ | $0.60 \pm 0.01$ | $4.19 \pm 0.23$ | |
| | GraFITi | $\underline{4.45 \pm 0.04}$ | $\underline{2.72 \pm 0.01}$ | $\underline{3.60 \pm 0.01}$ | $0.60 \pm 0.01$ | $4.25 \pm 0.04$ | |
| **ReIMTS** | +mTAN | $6.54 \pm 0.02$ | $3.49 \pm 0.01$ | $3.94 \pm 0.01$ | $0.96 \pm 0.01$ | $\mathbf{3.76 \pm 0.05}$ | ↑**20.6%** |
| | +GRU-D | $5.17 \pm 0.05$ | $3.85 \pm 0.12$ | $3.72 \pm 0.05$ | $1.39 \pm 0.02$ | $4.23 \pm 0.06$ | ↑**15.5%** |
| | +Raindrop | $5.52 \pm 0.05$ | $3.80 \pm 0.02$ | $3.74 \pm 0.01$ | $0.97 \pm 0.01$ | $4.36 \pm 0.11$ | ↑**8.4%** |
| | +PrimeNet | $5.17 \pm 0.03$ | $3.53 \pm 0.01$ | $3.67 \pm 0.01$ | $0.84 \pm 0.01$ | $4.29 \pm 0.08$ | ↑**53.8%** |
| | +TimeCHEAT | $4.75 \pm 0.01$ | $2.91 \pm 0.02$ | $3.55 \pm 0.01$ | $0.69 \pm 0.04$ | $3.91 \pm 0.05$ | ↑**3.2%** |
| | +GraFITi | $\mathbf{4.40 \pm 0.02}$ | $\mathbf{2.50 \pm 0.01}$ | $\mathbf{3.53 \pm 0.01}$ | $\mathbf{0.55 \pm 0.00}$ | $4.14 \pm 0.06$ | ↑**4.4%** |

## B.5 ADDITIONAL EFFICIENCY ANALYSIS

We provide additional efficiency comparisons on datasets MIMIC-IV, PhysioNet'12, Human Activity, and USHCN, using the same settings as in Table 3. Results are summarized in Figure 6. ReIMTS uses GraFITi as its backbones in these comparisons. As can be seen, on most datasets, our method ReIMTS uses significantly fewer GPU memory than other multi-scale IMTS methods, including Warpformer, HD-TTS, and Hi-Patch. The training speed is similar to other multi-scale methods, while ReIMTS can achieve better performance. ReIMTS also has comparable efficiency with the original model GraFITi.

## C BACKBONE DETAILS

We introduce how existing IMTS models servce as backbones in ReIMTS, with modifications potentially including reductions in hidden dimension size, layer count, or the removal of specific modules. Also, the structure of decoders and how backbones handle future timestamp queries and irregularities are also explained. We use the same strategy in the search of hyperparameters as compared baseline models, which aims to minimize the loss on validation sets. The number of layers is searched within 1, 2, 3, and 4, and the hidden dimension size is searched within 16, 32, 64, 128, 256, and 512. The number of scale levels used by ReIMTS are described in following paragraphs, which differs based on backbones. The time period lengths for all variants follow these settings: (1) MIMIC-III: 48 hours, 24 hours, 12 hours, and 6 hours; (2) MIMIC-IV: 48 hours, 24 hours, 12 hours, and 6 hours; (3) PhysioNet'12: 48 hours, 24 hours, 12 hours, and 6 hours; (4) Human Activity: 4000 milliseconds, 2000 milliseconds, 1000 milliseconds, and 500 milliseconds; (5) USHCN: 4 years, 2 years, 1 year, and 6 months. In the following descriptions, we use $L_S$ and $L_Q$ to denote the maximum number of observations in a univariate series in lookback and forecast window, respectively.

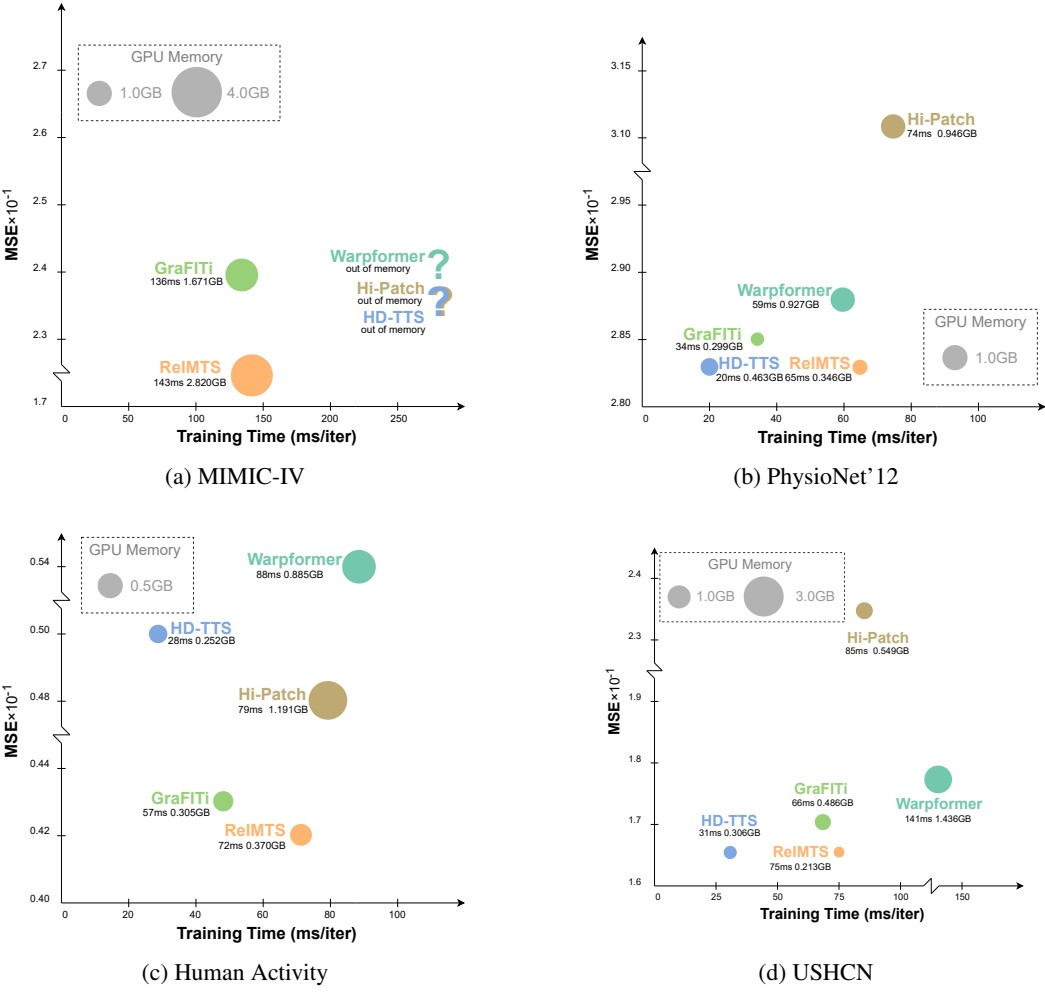

Figure 6: Efficiency comparisons on MIMIC-IV, PhysioNet'12, Human Activity, and USHCN with a batch size of 32. ReIMTS demonstrates significantly lower GPU memory usage than other multi-scale methods Warpformer, HD-TTS, and Hi-Patch on most datasets, while maintaining similar training time. Compared to the original backbone GraFITi, ReIMTS achieves comparable efficiency.

**ReIMTS+mTAN** uses mTAN (Shukla & Marlin, 2021a) as backbones. mTAN encodes input IMTS into temporal representations at a fixed set of reference points, described in Figure 2(b) of its paper. Therefore, it belongs to the case $\mathbf{E}^n = \mathbf{E}_{\text{time}}^n$ in Eq. 9, and we pass these temporal representations from top to bottom within the ReIMTS architecture to learn multi-scale temporal representations. For the decoder of mTAN, it consists of a GRU, a multi-head attention, and an MLP sequentially. It maps temporal representations of shape $L_S \times D$ into time series $L_Q \times V$. Future timestamps are used as queries in the multi-head attention, and the number of variables corresponds to the output dimension of the MLP. As for the hyperparameters, the hidden dimension size is set to 128, the number of reference points is 32, and the number of scale levels used by ReIMTS is 2 on all five datasets. The learning rate is $1 \times 10^{-3}$.

**ReIMTS+GRU-D** uses GRU-D (Che et al., 2018) as backbones. GRU-D encodes input IMTS into temporal representations for both lookback and future timestamps, as described in Eq.16 of its paper. Therefore, it belongs to the case $\mathbf{E}^n = \mathbf{E}_{\text{time}}^n$ in Eq. 9 and learns multi-scale temporal representations. For the decoder of GRU-D, it consists of an MLP maps temporal representations of shape $(L_S + L_Q) \times D$ into time series $(L_S + L_Q) \times V$. Therefore, future timestamps are included in the temporal representations, and the number of variables corresponds to the output dimension of the linear layer. To handle irregularities, GRU-D replaces padding values with the last observed values before processing through a GRU unit. After generating the predicted series, it uses a mask to retain only the observed and predicted values. As for the hyperparameters, since the original paper of GRU-D set the hidden dimension size to 100, we additionally include this setting during hyperparameter searching. The hidden dimension size is set to 100, 64, 64, 64, and 32 on dataset MIMIC-III, MIMIC-IV, PhysioNet'12, Human Activity, and USHCN, respectively. The number of scale levels used by ReIMTS is set to 2 on all datasets except MIMIC-III, which adopts 3 levels instead. The learning rate is $1 \times 10^{-3}$.

**ReIMTS+Raindrop** uses Raindrop (Zhang et al., 2022) as backbones. Raindrop encodes input IMTS into observation representations, as described in Eq.2 of its paper. Therefore, it belongs to the case $\mathbf{E}^n = \mathbf{E}_{\text{obs}}^n$ in Eq. 10 during implementation and learns multi-scale observational representations. Other modifications include reducing the number of propagation layers from two to one and removing the final transformer encoder layer. For the decoder of Raindrop, it is a linear layer that maps then rearranges variable representations of shape $V \times D$ into a time series with shape $L_Q \times V$. The variable IDs are learned within these representations, and the forecast length is the output dimension of the linear layer. To handle irregularities, it employs the mask on learned representations to retain only the positions corresponding to actual input observations. As for the hyperparameters, the number of layers used by Raindrop backbone is set to 2 on all five datasets. The hidden dimension size is set to 64, 1, 16, 64, and 16 on dataset MIMIC-III, MIMIC-IV, PhysioNet'12, Human Activity, and USHCN, respectively. The number of scale levels used by ReIMTS is set to 2 on all five datasets. The learning rate is $1 \times 10^{-3}$.

**ReIMTS+PrimeNet** uses PrimeNet (Chowdhury et al., 2023) as backbones. PrimeNet encodes input IMTS into temporal representations for both lookback and future timestamps, as described in Eq.2 in its paper. Therefore, it belongs to the case $\mathbf{E}^n = \mathbf{E}_{\text{time}}^n$ in Eq. 9 and learns multi-scale temporal representations. We disable the patch splitting operation in the original PrimeNet. For the decoder of PrimeNet, it is also an MLP maps temporal representations of shape $(L_S + L_Q) \times D$ into time series $(L_S + L_Q) \times V$. Future timestamps are included in the temporal representations, and the number of variables corresponds to the output dimension of the linear layer. To handle irregularities, it retains only the observed and predicted values after obtaining output series by applying the mask. As for the hyperparameters, the hidden dimension size is set to 256, 256, 256, 256, and 128 on dataset MIMIC-III, MIMIC-IV, PhysioNet'12, Human Activity, and USHCN, respectively. The number of scale levels used by ReIMTS is set to 2 on all five datasets. The learning rate is $1 \times 10^{-4}$.

**ReIMTS+TimeCHEAT** uses TimeCHEAT (Liu et al., 2025) as backbones. TimeCHEAT is based on GraFITi (Yalavarthi et al., 2024) and simultaneously learns temporal, variable, and observation representations, as described from Eq.3 to Eq.10 of its paper. We found that sharing variable embeddings across different scale levels is sufficient during our experiments. Therefore, it belongs to the case $\mathbf{E}^n = \mathbf{E}_{\text{var}}^n$ in Eq. 11 during implementation and learns multi-scale variable representations. The patch splitting operation in orginal TimeCHEAT is disabled, and the final transformer encoding

is discarded. Other settings remain the same. For the decoder of TimeCHEAT, it is a linear layer that decodes and squeezes representations of shape $(L_S + L_Q) \times V \times (3D)$ into time series $(L_S + L_Q) \times V$. These representations are obtained by concatenating temporal, variable, and observational representations along the hidden dimension, each repeated and expanded to shape $(L_S + L_Q) \times V \times D$. Therefore, future timestamps are included in the temporal representations, while variable IDs are included in the variable ones. To handle irregularities, it uses a bipartite graph approach like GraFITi, which removes all padding values and transform inputs into bipartite graphs. As for the hyperparameters, the number of layers is set to 2, 2, 4, 4, and 4 on dataset MIMIC-III, MIMIC-IV, PhysioNet'12, Human Activity, and USHCN, respectively. The hidden dimension size is set to 128, 64, 32, 64, and 64 on dataset MIMIC-III, MIMIC-IV, PhysioNet'12, Human Activity, and USHCN, respectively. The number of scale levels used by ReIMTS is set to 2 on all five datasets. The learning rate is $1 \times 10^{-3}$.

**ReIMTS+GraFITi** uses GraFITi (Yalavarthi et al., 2024) as backbones. GraFITi simultaneously learns temporal, variable, and observation representations, as described from Eq.11 to Eq.13 of its paper. During our implementation, we only transfer variable representations across different scale levels. Therefore, it belongs to the case $\mathbf{E}^n = \mathbf{E}_{var}^n$ in Eq. 11 during implementation and learns multi-scale variable representations. For the decoder of GraFITi, please refer to the descriptions of TimeCHEAT's decoder above. To handle irregularities, it uses a bipartite graph approach that removes all padding values and transform inputs into bipartite graphs. As for the hyperparameters, the number of layers is set to 4, 1, 1, 2, and 1 on dataset MIMIC-III, MIMIC-IV, PhysioNet'12, Human Activity, and USHCN, respectively. The hidden dimension size is set to 128, 32, 256, 128, and 32 on dataset MIMIC-III, MIMIC-IV, PhysioNet'12, Human Activity, and USHCN, respectively. The number of scale levels used by ReIMTS is set to 2 on all datasets except PhysioNet'12 and MIMIC-III, which adopt 3 levels instead. The learning rate is $1 \times 10^{-3}$.

## D VISUALIZATION

### D.1 FORECASTING RESULTS

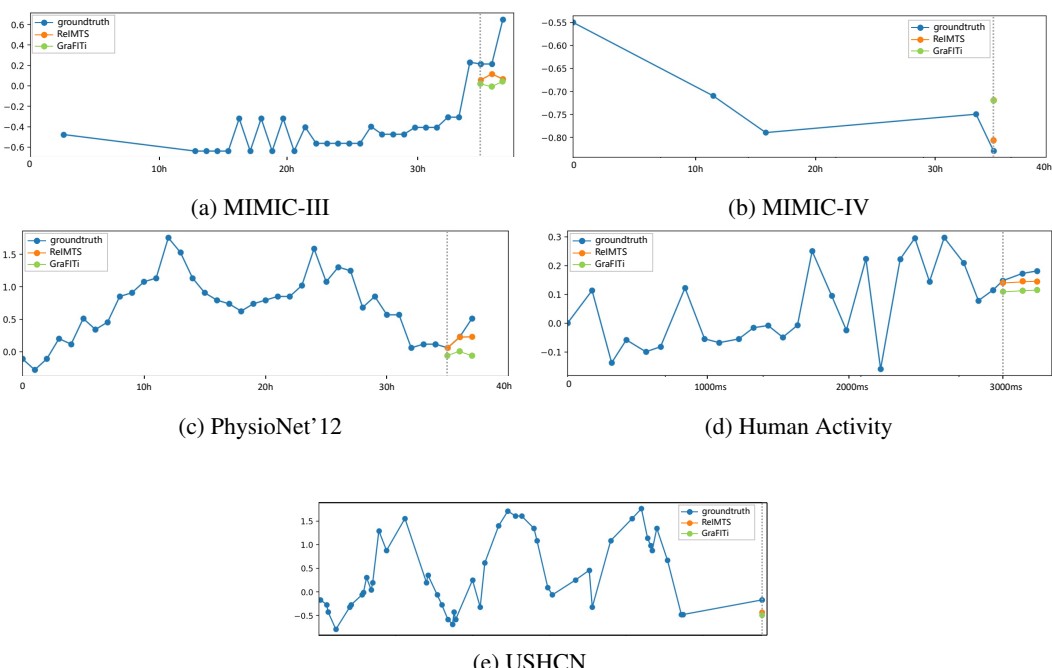

(a) MIMIC-III

(b) MIMIC-IV

(c) PhysioNet'12

(d) Human Activity

(e) USHCN

Figure 7: Visualization of forecast results on all five datasets, under the same settings as in Table 2. Compared to original backbone GraFITi (in green), ReIMTS (in orange) is closer to the ground truth (in blue).

We visualize the forecasting results of our proposed ReIMTS when using GraFITi as its backbone, and also ones from the original GraFITi model, as depicted in Figure 7. The forecast settings are exactly the same as in Table 2, which means the lookback window length is set to 36 hours for MIMIC-III, MIMIC-IV, and PhysioNet'12, 3000 milliseconds for Human Activity, and 3 years for USHCN. The goal is to predict next 300ms for Human Activity, and 3 timestamps for other datasets. As can be seen, ReIMTS is closer to the ground truth value.

## D.2 MULTI-SCALE REPRESENTATIONS

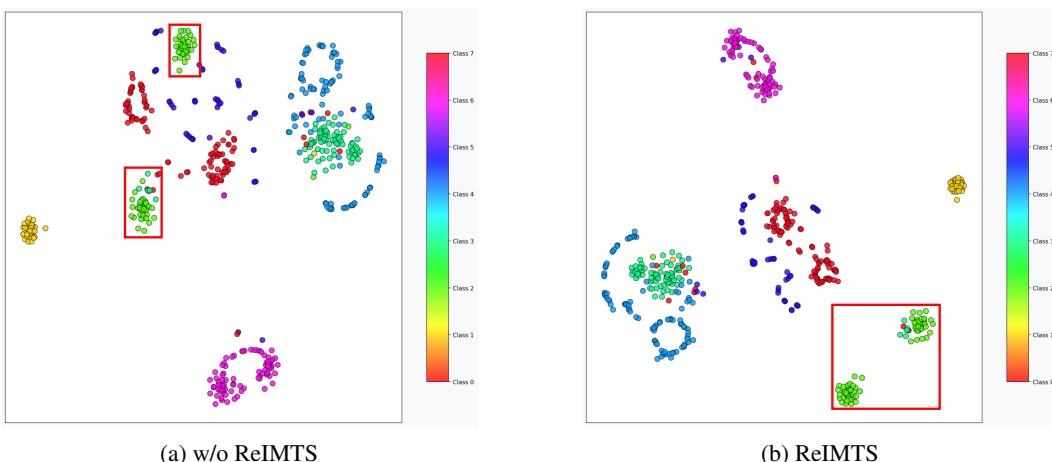

(a) w/o ReIMTS                                        (b) ReIMTS

Figure 8: The t-SNE visualization of multi-scale representations in ReIMTS+GRU-D on classification dataset PAM. Red boxs indicate the region of interest, which mainly contain samples from class 2. (a) Without ReIMTS, samples from class 2 are very close to ones from class 5 and 7. (b) After applying ReIMTS, samples from class 2 are more distant from others, leading to improved performance. It should be noted that class 3 and class 2 have similar colors, where samples from class 3 are mostly outside of the region of interest.

Since it's hard to visualize the relationships between learned representations and forecasting accuracy, we train ReIMTS+GRU-D on classification task instead for visualization, as depicted in Figure 8 where colored points represent test set samples. GraFITi is not originally designed for classification task, so we choose the classic and widely acknowledged GRU-D as backbones for ReIMTS. Widely studied dataset PAM (Reiss & Stricker, 2012) is used, which contains 8 class labels. We follow the preprocessing scripts of Raindrop (Zhang et al., 2022) and split the train/val/test sets adhere to ratio 8:1:1. Other training protocols are the same as forecasting training in Table 3. As can be seen in Figure 8 (a), when learning without ReIMTS, samples from class 2 are very close to ones from class 5 and 7. The test set performance is 84.14% in precision, 78.04% in recall, and 75.03% in F1. After applying ReIMTS in Figure 8 (b), samples from class 2 are more distant from others multi-scale representations. The test set performance is improved to 86.74% in precision, 84.04% in recall, and 82.44% in F1.

## E   BASELINE DETAILS

We briefly introduce each baseline model along with their key hyperparameter settings here. The search of hyperparameters aims to minimize the loss on validation sets. The number of layers is searched within 1, 2, 3, and 4, the hidden dimension size is searched within 16, 32, 64, 128, 256, and 512. Unless otherwise specified, we use a batch size of 16 for USHCN, and 32 for others. However, if the model cannot be trained using 24 GB of GPU memory, we recursively halve the batch size until training is possible. Number of epochs, early stopping patience, random seeds, and learning rates have been described in Section 5.1.3. We aim to use the same hyperparameters for learning rate and special loss functions, as specified in their original papers and codes, whenever available. For all classification models, we replace the final softmax layer with a linear layer to enable forecasting.

E.1 METHODS FOR MTS

**MOIRAI** (Woo et al., 2024) is a pretraining model for time series forecasting. We use the small version of the provided pretrained configurations and weights, comprising 6 layers with hidden dimension 384. We finetune the model on IMTS datasets with learning rate of $1 \times 10^{-4}$.

**Ada-MSHyper** (Shang et al., 2024) uses hypergraphs for temporal multi-scale learning in MTS forecasting. The window size for multiscale is 4. The learning rate is $1 \times 10^{-3}$. We also use its node and hyperedge constrainted loss function for training. The number of layers is set to 1, 1, 2, 1, and 1 on dataset MIMIC-III, MIMIC-IV, PhysioNet'12, Human Activity, and USHCN, respectively. The hidden dimension size is set to 32, 256, 256, 512, and 64 on dataset MIMIC-III, MIMIC-IV, PhysioNet'12, Human Activity, and USHCN, respectively.

**Autoformer** (Wu et al., 2021) is a transformer variant with auto-correlation decomposition for MTS forecasting. The attention factor is 3. The learning rate is $1 \times 10^{-3}$. The number of layers is set to 4, 2, 1, 1, and 1 on dataset MIMIC-III, MIMIC-IV, PhysioNet'12, Human Activity, and USHCN, respectively. The hidden dimension size is set to 32, 32, 64, 512, and 128 on dataset MIMIC-III, MIMIC-IV, PhysioNet'12, Human Activity, and USHCN, respectively.

**Scaleformer** (Shabani et al., 2023) is a coarse-to-fine multi-scale framework for transformer models in MTS forecasting. We adopt its best-performing backbone evaluated in original paper, Autoformer, in our benchmark. The scale factor is 2. The attention factor is 3. The number of encoder layers is 4 on dataset USHCN, and 2 on the rest. The number of decoder layers is 1. The hidden dimension size is set to 128, 256, 64, 256, and 16 on dataset MIMIC-III, MIMIC-IV, PhysioNet'12, Human Activity, and USHCN, respectively. The learning rate is $1 \times 10^{-3}$.

**TimesNet** (Wu et al., 2023) uses 2-D variation modeling for MTS analysis. The attention factor is 3. The dimension for FCN is 32. The learning rate is $1 \times 10^{-3}$. The number of encoder layers is set to 2. The hidden dimension size is set to 256, 128, 128, 32, and 128 on dataset MIMIC-III, MIMIC-IV, PhysioNet'12, Human Activity, and USHCN, respectively.

**NHITS** (Challu et al., 2023) is a multi-scale model with hierarchical interpolation technique for MTS forecasting. The learning rate is $1 \times 10^{-3}$. The hidden dimension size is set to 64, 32, 32, 512, and 128 on dataset MIMIC-III, MIMIC-IV, PhysioNet'12, Human Activity, and USHCN, respectively.

**PatchTST** (Nie et al., 2023) leverages patching in transformer for MTS forecasting. The patch lengths are 12, 90, 6, 300 and 10 for MIMIC-III, MIMIC-IV, PhysioNet'12, Human Activity, and USHCN, respectively. The number of encoder layers is 8 on USHCN, and 3 on the rest. The attention factor is 3. The number of heads in attention is 8 on USHCN, and 16 on the rest. The learning rate is $1 \times 10^{-4}$. The hidden dimension size is set to 128, 128, 256, 512, and 32 on dataset MIMIC-III, MIMIC-IV, PhysioNet'12, Human Activity, and USHCN, respectively.

**Leddam** (Yu et al., 2024) uses learnable seasonal-trend decomposition for MTS forecasting. The learning rate is $1 \times 10^{-3}$. The number of layers is set to 2, 4, 2, 1, and 4 on dataset MIMIC-III, MIMIC-IV, PhysioNet'12, Human Activity, and USHCN, respectively. The hidden dimension size is set to 32, 32, 128, 512, and 64 on dataset MIMIC-III, MIMIC-IV, PhysioNet'12, Human Activity, and USHCN, respectively.

**Pathformer** (Chen et al., 2024) is a multi-scale transformer with multi patch size aggregation. Since Pathformer does not support splitting irregular time series into patches of different lengths, the number of layers is fixed as 1. The hidden dimension size is set to 32, 16, 32, 16, and 16 on dataset MIMIC-III, MIMIC-IV, PhysioNet'12, Human Activity, and USHCN, respectively. The patch lengths are 24, 720, 12, 1000, and 50 for MIMIC-III, MIMIC-IV, PhysioNet'12, Human Activity, and USHCN, respectively. The learning rate is $1 \times 10^{-3}$.

**Crossformer** (Zhang & Yan, 2023) learns cross-dimensional dependencies for MTS forecasting. The segment lengths are 12, 360, 6, 300, and 12 for MIMIC-III, MIMIC-IV, PhysioNet'12, Human

Activity, and USHCN respectively. The learning rate is $1 \times 10^{-3}$. The number of encoder layers is set to 8 on USHCN, and 2 on the rest. The hidden dimension size is set to 32, 32, 128, 512, and 64 on dataset MIMIC-III, MIMIC-IV, PhysioNet'12, Human Activity, and USHCN, respectively.

**TimeMixer** (Wang et al., 2024a) uses seasonal-trend decomposition at each sampling scale level. The down sampling windows are 24, 720, 12, 300, and 50 for MIMIC-III, MIMIC-IV, PhysioNet'12, Human Activity, and USHCN, respectively. The number of encoder layer is 8 on USHCN, and 3 on the rest. The dimension for feed-forward layer is 16 on USHCN, and 32 on the rest. The learning rate is $1 \times 10^{-2}$. The number of downsampling layers is set to 1. The hidden dimension size is set to 32, 32, 64, 16, and 256 on dataset MIMIC-III, MIMIC-IV, PhysioNet'12, Human Activity, and USHCN, respectively.

### E.2 METHODS FOR IMTS

**PrimeNet** (Chowdhury et al., 2023) is an IMTS pretraining model. Since the provided weights are specific to datasets with 41 variables, we retrain the model on all datasets. The patch lengths are 12, 180, 6, 300, and 10 for MIMIC-III, MIMIC-IV, PhysioNet'12, Human Activity, and USHCN respectively. The number of heads in attention is 1. The learning rate is $1 \times 10^{-4}$. The hidden dimension size is set to 32, 32, 128, 128, and 64 on dataset MIMIC-III, MIMIC-IV, PhysioNet'12, Human Activity, and USHCN, respectively.

**SeFT** (Horn et al., 2020) is a set-based method for IMTS classification. The dropout rate is 0.1. The learning rate is $1 \times 10^{-3}$. The number of layers is set to 2, 2, 4, 4, and 4 on dataset MIMIC-III, MIMIC-IV, PhysioNet'12, Human Activity, and USHCN, respectively. The hidden dimension size is set to 256, 64, 32, 128, and 16 on dataset MIMIC-III, MIMIC-IV, PhysioNet'12, Human Activity, and USHCN, respectively.

**mTAN** (Shukla & Marlin, 2021a) converts IMTS to reference points for IMTS classification. The number of reference points is 32 on MIMIC-III, 128 on USHCN, and 8 on the rest datasets. The hidden dimension size is set to 32, 64, 64, 64, and 256 on dataset MIMIC-III, MIMIC-IV, PhysioNet'12, Human Activity, and USHCN, respectively. The learning rate is $1 \times 10^{-3}$.

**NeuralFlows** (Biloš et al., 2021) is an efficient alternative to Neural ODE in IMTS analysis. The number of flow layers is 2. The latent dimension is 20. The hidden dimension for time is 8. The number of hidden layers is 3. The learning rate is $1 \times 10^{-3}$.

**CRU** (Schirmer et al., 2022) uses continuous recurrent units for IMTS analysis. The hidden dimension is 20. The learning rate is $1 \times 10^{-3}$.

**TimeCHEAT** (Liu et al., 2025) uses channel-dependent within patches and channel-independent among patches. The patch lengths are 12, 360, 6, 1000, and 50 for MIMIC-III, MIMIC-IV, PhysioNet'12, Human Activity, and USHCN respectively. The number of attention heads is 8 on USHCN, and 4 on the rest. The learning rate is $1 \times 10^{-3}$. The number of layers is set to 1, 1, 1, 2, and 1 on dataset MIMIC-III, MIMIC-IV, PhysioNet'12, Human Activity, and USHCN, respectively. The hidden dimension size is set to 32, 32, 32, 64, and 128 on dataset MIMIC-III, MIMIC-IV, PhysioNet'12, Human Activity, and USHCN, respectively.

**GNeuralFlow** (Mercatali et al., 2024) enhances NeuralFlows with graph neural networks for IMTS analysis. The flow model uses ResNet. The number of flow layers is 2. The latent dimension for input is 20. The latent dimension for time is 8. The number of hidden layers is 3. The learning rate is $1 \times 10^{-3}$.

**GRU-D** (Che et al., 2018) adapts GRUs for IMTS classification. The learning rate is $1 \times 10^{-3}$. Since the original paper of GRU-D set the hidden dimension size to 100, we additionally include this setting during hyperparameter searching. The hidden dimension size is set to 100, 32, 100, 100, and 128 on dataset MIMIC-III, MIMIC-IV, PhysioNet'12, Human Activity, and USHCN, respectively.

**Raindrop** (Zhang et al., 2022) models time-varying variable dependencies for IMTS classification. The learning rate is $1 \times 10^{-4}$. The number of layers is set to 1, 2, 1, 1, and 2 on dataset MIMIC-III, MIMIC-IV, PhysioNet'12, Human Activity, and USHCN, respectively. The hidden dimension size is set to 32.

**tPatchGNN** (Zhang et al., 2024) processes IMTS into patches and use graph neural networks for IMTS forecasting. The patch lengths are 12, 360, 6, 300, and 10 for MIMIC-III, MIMIC-IV, PhysioNet'12, Human Activity, and USHCN, respectively. The number of heads in attention is 4 on USHCN, and 1 on the rest. The learning rate is $1 \times 10^{-3}$. The number of layers is set to 4, 2, 4, 4, and 3 on dataset MIMIC-III, MIMIC-IV, PhysioNet'12, Human Activity, and USHCN, respectively. The hidden dimension size is set to 64, 32, 64, 64, and 32 on dataset MIMIC-III, MIMIC-IV, PhysioNet'12, Human Activity, and USHCN, respectively.

**Hi-Patch** (Luo et al., 2025) implements hierarchical graph fusion inside backbone. The patch lengths are 12, 90, 6, 500, and 6 for MIMIC-III, MIMIC-IV, PhysioNet'12, Human Activity, and USHCN, respectively. The number of attention heads is 8 on USHCN and 1 on the rest. The learning rate is $1 \times 10^{-3}$. The number of layers is set to 4, 1, 4, 4, and 4 on dataset MIMIC-III, MIMIC-IV, PhysioNet'12, Human Activity, and USHCN, respectively. The hidden dimension size is set to 32, 64, 32, 128, and 32 on dataset MIMIC-III, MIMIC-IV, PhysioNet'12, Human Activity, and USHCN, respectively.

**Warpformer** (Zhang et al., 2023a) uses warping for multiscale modeling in IMTS classification. The number of heads is 1. The dropout rate is 0. The learning rate is $1 \times 10^{-3}$. The number of layers is set to 4, 1, 1, 3, and 1 on dataset MIMIC-III, MIMIC-IV, PhysioNet'12, Human Activity, and USHCN, respectively. The hidden dimension size is set to 64.

**HD-TTS** (Marisca et al., 2024) implements both variable and temporal multi-scale inside backbone. The number of rnn layers is 4. The number of pooling layers is 1. The learning rate is $1 \times 10^{-3}$. The hidden dimension size is 64.

**GraFITi** (Yalavarthi et al., 2024) uses bipartite graphs for IMTS forecasting. The number of heads in attention is 4. The learning rate is $1 \times 10^{-3}$. The number of layers is set to 3, 1, 2, 4, and 1 on dataset MIMIC-III, MIMIC-IV, PhysioNet'12, Human Activity, and USHCN, respectively. The hidden dimension size is set to 256, 32, 32, 32, and 128 on dataset MIMIC-III, MIMIC-IV, PhysioNet'12, Human Activity, and USHCN, respectively.

## F THE USE OF LARGE LANGUAGE MODELS

We use Large Language Models (LLMs) to polish writing only. Specifically, we write all contents ourselves, then use LLMs to check for any possible grammar mistakes or incorrect usage of words. We have not used LLMs for any other purpose.

