# OpenReview forum: "Learning Recursive Multi-Scale Representations for Irregular Multivariate Time Series Forecasting"
_ICLR.cc/2026/Conference — ICLR 2026 Poster_

### Official Review · Reviewer_4Jcx · 2025-10-27

**Soundness:** 3
**Presentation:** 3
**Contribution:** 3
**Rating:** 6
**Confidence:** 4

**Summary:**

The paper addresses the valuable problem of learning multi-scale information from irregular time series. ReIMTS, a multi-scale method based on recursive splitting followed by concatenation, is proposed. Forecasting experiments demonstrate its competitive performance across settings. While the idea is overall well-motivated and novel, more explanatory experiments could further strengthen the claims.

**Strengths:**

1.The paper is well-motivated, and the proposed ReIMTS is an interesting and easily understandable solution to the problem.

2.Benchmark experiments are extensive for the IMTS forecasting task, and the corresponding codes are available.

**Weaknesses:**

1.Although Figure 1 illustrates a sampling pattern present in the dataset, the term "sampling pattern" is rarely used in existing works. More careful explanations of this concept are needed.

2.At line 187, the paper states that $L^n$ and $T^n$ are distinct, with the difference lying in the presence of time units. The notation causes some confusion during reading, raising questions about whether it is necessary to distinguish them using different symbols.

3.Although ReIMTS+GraFITi appears to have good efficiency compared to other multi-scale methods, its training time (66ms) is double that of GraFITi (33ms). This raises doubts about the efficiency of ReIMTS when applied to other backbones.


4.The paper is entitled “Learning... Representations...”, which implies that ReIMTS functions as a representation learning booster. While the benchmark experiments on the forecasting task are extensive, additional experiments on the classification task could make the claim more persuasive.

**Questions:**

1.As mentioned in Weakness 2, is it really necessary to distinguish between $ L^n $ and $ T^n $, two symbols that differ only in the presence of time units? For example, in line 203, why is $ e^1_{\text{time},0:T^1} $ using the symbol with time units, when it actually belongs to $ R^{L^1 \times d_{\text{model}}} $, which uses the symbol without time units?

2.As mentioned in weakness 3, how is the efficiency of ReIMTS when applied to other backbones, like PrimeNet[1] and mTAN[2]?

3.In Section 4.2, the paper introduces a representation fusion method that employs a scoring linear layer to assign weights to global representations. Could the authors provide further insights into this design to clarify how different representations are combined? Specifically, does ReIMTS tend to prioritize the use of local representations or global ones?

4.Representations learned in the classification task are plotted in Figure 8. However, benchmark comparisons have not included metrics from classification. How does ReIMTS perform under other downstream tasks, such as classification?


5.The paper's experimental settings follow those from HyperIMTS[3], while the HyperIMTS model is not included in the benchmark comparisons. Why is it not included? Can ReIMTS work with HyperIMTS?

[1] R. R. Chowdhury, J. Li, X. Zhang, D. Hong, R. K. Gupta, and J. Shang; “PrimeNet: Pre-training for Irregular Multivariate Time Series”; AAAI 2023.

[2] S. N. Shukla and B. Marlin; “Multi-Time Attention Networks for Irregularly Sampled Time Series”; ICLR 2021.

[3] B. Li, Y. Luo, Z. Liu, J. Zheng, J. Lv, and Q. Ma; “HyperIMTS: Hypergraph Neural Network for Irregular Multivariate Time Series Forecasting”; ICML 2025.

---

> ### Author Response · Authors · 2025-11-24
>
> We thank the reviewer for finding our work **well-motivated and interesting** as strengths!
> **We have updated our manuscript**, and your concerns are addressed as follows:
>
> ## W1. Sampling pattern explanation
>
> **A1**: We apologize for any confusion regarding the term "sampling pattern", and we provide its definition as:
> > A **sampling pattern** is the ordered set of time instants $\{t_{1},t_{2},\dots ,t_{n}\}$ at which a time‑varying quantity $x(t)$ is observed. It determines the temporal spacing between samples, which may be irregular.
>
> ## W2 & Q1. Distinguish $L^n$ and $T^n$
>
> **A2**:
> ||$L^1$|$T^1$|
> |---|---|---|
> |meaning|max number of observations along time|time periods|
> |unit|-|minute/hour/year/etc.
>
> Since IMTS have varying sampling rates, the equation $L^1=\text{sampling rate}\times T^1$ does not hold.
> Therefore, it is necessary to use two separate symbols to distinguish.
>
> ## W3 & Q2. Efficiency when applied to other backbones
>
> **A3**: We present a comparison of ReIMTS training time (ms) across different backbones:
> ||MIMIC-III|MIMIC-IV|P12|USHCN|Human Activity|
> |---|---|---|---|---|---|
> |PrimeNet|**32**|ME|24|81|63|
> |**+ReIMTS**|41|ME|24|**72**|63|
> |TimeCHEAT|381|446|240|**136**|145|
> |**+ReIMTS**|**357**|ME|**184**|170|**135**|
> |GRU-D|**60**|**563**|**37**|**267**|**127**|
> |**+ReIMTS**|126|949|53|414|169|
> |mTAN|37|88|**15**|**50**|54|
> |**+ReIMTS**|**19**|**37**|17|55|**51**|
> |Raindrop|75|ME|**53**|**80**|ME|
> |**+ReIMTS**|ME|304|81|119|340|
>
> As shown above, **ReIMTS can achieve comparable or even faster speed**.
>
> ## W4 & Q4. Classification comparisons
>
> **A4**: We present the classification results on **P12 [1], P19 [2], and PAM [3]** following the data preprocessing setup in Raindrop [4].
> The standard deviations are omitted to save space.
> Metrics are expressed in percent (\%).
>
> ||P12 (AUROC)|P12 (AUPRC)|P19 (AUROC)|P19 (AUPRC)|PAM (Precision)|PAM (Recall)
> |---|---|---|---|---|---|---|
> |PrimeNet|85.52|52.63|83.21|29.15|72.89|70.44|
> |**+ReIMTS**|**85.85**|**54.63**|**85.52**|**30.53**|**75.91**|**71.34**|
> |TimeCHEAT|52.13|16.28|48.47|3.31|66.10|59.18|
> |**+ReIMTS**|**86.2**|**56.46**|**85.24**|**44.27**|**87.6**|**86.23**|
> |GraFITi|**87.83**|58.85|82.2|43.66|76.79|73.91|
> |**+ReIMTS**|87.57|**59.76**|**86.52**|**49.02**|**82.88**|**82.64**|
> |GRU-D|78.83|45.24|84.68|**47.25**|84.57|75.27|
> |**+ReIMTS**|**84.71**|**53.57**|**84.90**|41.46|**85.29**|**75.40**|
> |mTAN|84.26|**53.76**|79.55 |25.15|52.70|54.01|
> |**+ReIMTS**|**85.91**|52.66|**82.82**|**25.27**|**65.55**|**54.20**|
> |Raindrop|83.87|48.50|76.63|36.21|81.00|76.85|
> |**+ReIMTS**|**84.54**|**49.52**|**77.25**|**38.43**|**85.24**|**81.27**|
> As evident, ReIMTS **improves performance in almost all settings**.
>
> ## Q3. Representation fusion explanation
>
> **A5**: In short, we found that **global representations tend to be important in representation fusion when the number of observations is insufficient**.
> The conclusion is supported by **Spearman correlation**.
> We use ReIMTS+GraFITi on dataset MIMIC-IV to illustrate:
> - Calculate the number of observed values $Y_{\text{sub}}$ in each subsample at scale level 2, given a time span of 24 hours (half of the original 48 hours at scale level 1).
> - Extract the sum of scores $\alpha_\text{sub}$ for each subsample during representation fusion.
> - Spearman correlation between $Y_{\text{sub}}$ and $\alpha_\text{sub}$ is calculated as **-0.65**. Values closer to -1 indicate a stronger negative association
>
> ## Q5. ReIMTS+HyperIMTS
>
> **A6**:
> Apart from the results list in the original paper of HyperIMTS, we also apply the same hyperparameter searching process described in Appendix E of revised manuscript, and report both the original and tuned results (MSE$\times10^{-1}$):
> ||MIMIC-III|MIMIC-IV|P12|Human Activity|USHCN|
> |-----|---|---|---|---|---|
> |HyperIMTS(original)|4.26±0.02|2.17±0.01|3.00±0.00|0.42±0.02|1.74±0.08|
> |HyperIMTS(tuned)|**3.67±0.02**|2.17±0.01|2.85±0.00|0.42±0.02|1.74±0.08|
> |ReIMTS+HyperIMTS|4.16±0.02|**1.86±0.03**|**2.79±0.00**|**0.42±0.00**|**1.30±0.07**|
>
> As can be seen, **ReIMTS improves the forecasting performance of HyperIMTS on 4 of 5 datasets**.
>
> [1] Silva, I., et al; "Predicting In-Hospital Mortality of ICU Patients: The PhysioNet/Computing in Cardiology Challenge 2012"; Comput Cardiol (2010)
>
> [2] Reyna, M.A. et al; "Early Prediction of Sepsis From Clinical Data: The PhysioNet/Computing in Cardiology Challenge 2019", Critical Care Medicine, 48(2), p. 210.
>
> [3] Reiss, A. and Stricker, D; "Introducing a New Benchmarked Dataset for Activity Monitoring", ISWC 2012
>
> [4] Zhang, X. et al; "Graph-Guided Network for Irregularly Sampled Multivariate Time Series"; ICLR 2022
>
> [5] Li, B., et al; "HyperIMTS: Hypergraph Neural Network for Irregular Multivariate Time Series Forecasting"; ICML 2025
>
> ---
> **Thanks again for your thorough review and looking forward to your reply!**

---

> > ### Comment · Reviewer_4Jcx · 2025-11-27
> > **Thank you for your detailed reply. Your response has effectively resolved my issue.**
> >
> > Thank you for your detailed reply. Your response has effectively resolved my issue. I have decided to revise my score to support the acceptance of this paper. Good luck！

---

> > > ### Author Response · Authors · 2025-11-27
> > >
> > > Thank you for taking the time to reconsider the review and for your helpful feedback. We sincerely appreciate your effort and consideration!

---

### Official Review · Reviewer_rfED · 2025-10-31

**Soundness:** 3
**Presentation:** 2
**Contribution:** 3
**Rating:** 4
**Confidence:** 3

**Summary:**

This paper proposes ReIMTS, a recursive multi-scale modeling approach for irregular multivariate time series (IMTS) forecasting, which splits samples recursively and employs irregularity-aware representation fusion mechanism

**Strengths:**

1. The motivation is clear, and the method is novel.
2. Experimental evaluation is comprehensive.
3. The method is easily adaptable to different models.

**Weaknesses:**

1. The notation and equations are not professional and can be largely improved. This paper uses a lot of double subscripts/superscript, and very long subscripts/superscripts, especially in Sec 4.1, from Eq. 2 to Eq. 6. I suggest the author reduce the length of Sec 4.1, and move some interesting experimental results or useful method comparison (Fig. 6) from the supplementary to the main paper.
2. Potential comparison fairness issues:
 - Backbone models are modified when integrated into ReIMTS (layers reduced, dimensions changed - Appendix B), unclear if these modifications could account for some improvements
 - Is there a difference in hyperparameter settings used for ReIMTS variants vs. baselines?
3. (minor) This paper could benefit from more discussion on the results. Table 2 shows that the performance improvement achieved by using different backbones is quite different, 62.3% vs. 9.9%. This is somewhat unusual and could benefit from an in-depth analysis.
4. (minor) Increased memory usage. Figure 3 shows that the available memory has increased by more than 50%(0.390 vs 0.598).

**Questions:**

I am happy to raise my rating of the paper if the author can address my concerns about **1) notations/equations** and **2) fairness**.

---

> ### Author Response · Authors · 2025-11-24
>
> We thank the reviewer for acknowledging the **clarity, novelty, adaptability, and comprehensive evaluation** of our work!
> We address your concerns as follows:
>
> ## W1. Notation and equation improvements
> **A1**: We sincerely thank the reviewer for the constructive comments!
> **The revised content has been highlighted in blue in the updated manuscript:**
>
> ### Notation overhaul
> 1. Use consistent typography: (1) Scalars: italic, both lowercase and uppercase; (2) Matrices: bold, lowercase; (3) A set of matrices/tuples: bold, uppercase; (4) Functions/Modules/Loss: calligraphic.
> 2. Avoid prime and double-prime notation.
> 3. Remove long index subscripts, and use sets to define index ranges.
> 4. Separate the shapes of tensors from definitions.
>
> ### Section restructuring and figure relocation
> 1. Section 4.1 contains essential equations only; the rest appear in Appendix A.
> 2. Figure 6 in original submission has been moved to Section 4.4.
>
> ## W2. Comparison fairness
>
> **A2**:
> We have paid great attention to comparison fairness:
>
> |Concerns|Answer|
> |---|---|
> |Hyperparameter differences|As noted in Appendix C of revised manuscript, we **use the same hyperparameter search process for ReIMTS**, because structural changes can shift the optimal backbones settings. Accordingly, hyperparameters may differ based on the search outcomes, **all listed in Appendices C and E**.|
> |Backbone modifications|Changes of number of layers and hidden size arise from **differences in hyperparameter search results**.|
>
> To justify the need for tuning, **we provide results using identical hyperparameters for ReIMTS and GraFITi [1], where no modifications are made to the backbone.**
> The experimental settings match Table 2.
> The standard deviations are omitted to save space in this comment.
>
> **Apply ReIMTS's search results to both models:**
> ||MIMIC-III|MIMIC-IV|P12|Human Activity|USHCN|
> |---|---|---|---|---|---|
> |GraFITi (not tuned)|3.72|2.39|2.85|0.49|1.84|
> |+ReIMTS (tuned)|**3.66**|**1.79**|**2.83**|**0.42**|**1.23**|
>
> **Apply GraFITi's search results to both models:**
> ||MIMIC-III|MIMIC-IV|P12|Human Activity|USHCN|
> |---|---|---|---|---|---|
> |GraFITi (tuned)|**3.70**|2.39|2.85|**0.43**|1.59|
> |+ReIMTS (not tuned)|4.68|**1.83**|**2.83**|0.57|**1.36**|
>
> **Fixed to 2 layers and 128 hidden size:**
> ||MIMIC-III|MIMIC-IV|P12|Human Activity|USHCN|
> |---|---|---|---|---|---|
> |GraFITi (fixed)|4.53|2.45|3.06|0.44|2.03|
> |+ReIMTS (fixed)|**4.48**|**2.01**|**2.94**|**0.42**|**1.23**|
>
> As can be seen, **only proper hyperparameter searches revealed ReIMTS and baseline's full potential**.
>
> ## W3. Performance improvements
>
> **A3**: We discuss from two perspectives:
>
> ### Backbone characteristic perspective
>
> We think **backbones that require more samples to converge gain the most from ReIMTS**, like PrimeNet:
> 1. **Recursive splitting increases the number of training subsamples across different time periods.**
>
>     The process acts like *a form of data augmentation*, encouraging backbones at different scales to learn from distinct “receptive fields” (i.e., time periods).
> 2. **PrimeNet, as a pre-training model, benefits from more samples generated by ReIMTS.**
>
>     It is built upon BERT, the well-known pre-trained language model.
>
> ### Dataset characteristic perspective
> In our **official comment A2 to Reviewer MQ28** above, we conclude that our method tend to excel at forecasting time series with *longer max length along time*.
> The recursive splitting divides long IMTS sample into shorter subsamples, which *facilitates local dependency learning*.
>
> ## W4. Memory usage
> **A4: The GPU memory usage mostly depends on the hyperparameter search results that achieve optimal performance**.
> We provide the statistics for the rest ReIMTS variants below, following settings in Figure 4.
> In short, *ReIMTS uses even less GPU memory in 12 of 25 comparisons*.
> Memory is measured in GB, and "ME" indicates memory error at batch size 32.
>
> |Model|MIMIC-III|MIMIC-IV|P12|Human Activity|USHCN
> |---|---|---|---|---|---|
> |PrimeNet|**0.785**|ME|**0.357**|**0.527**|1.165|
> |**+ReIMTS**|1.302|ME|0.437|0.600|**0.753**|
> |mTAN|1.291|3.350|0.117|0.317|0.215|
> |**+ReIMTS**|**0.330**|**0.788**|**0.083**|**0.124**|**0.113**|
> |TimeCHEAT|**4.676**|**5.659**|**1.524**|0.733|**0.224**|
> |**+ReIMTS**|7.978|ME|2.161|**0.487**|1.327|
> |GRU-D|**0.038**|**0.190**|**0.019**|**0.033**|0.163|
> |**+ReIMTS**|0.111|0.397|0.030|0.042|**0.056**|
> |Raindrop|ME|ME|0.968|**1.275**|8.486|
> |**+ReIMTS**|**1.409**|**3.928**|**0.469**|3.643|**1.619**|
>
> [1] Yalavarthi, V. K., et al; "GraFITi: Graphs for Forecasting Irregularly Sampled Time Series"; AAAI 2024
>
> [2] Chowdhury, R. R., et al; "PrimeNet: Pre-training for Irregular Multivariate Time Series"; AAAI 2023
>
> [3] Che, Z., et al; "Recurrent Neural Networks for Multivariate Time Series with Missing Values"; Sci Rep 2018
>
> ---
> **Thanks again for your valuable feedback!
> We hope that the revisions meet your expectations :).**

---

> ### Author Response · Authors · 2025-11-27
> **Gentle Follow-up on Author Response**
>
> Dear Reviewer rfED,
>
> I hope you are doing well. As the discussion period is nearing its end, we wanted to kindly check whether our author response has addressed your concerns.
>
> If there are any remaining issues that would benefit from additional clarification, we are happy to provide further details.
>
> Thank you very much for your time and thoughtful review.

---

### Official Review · Reviewer_XY3z · 2025-11-01

**Soundness:** 3
**Presentation:** 3
**Contribution:** 3
**Rating:** 6
**Confidence:** 4

**Summary:**

This paper proposes ReIMTS, a recursive multi-scale modeling framework for Irregular Multivariate Time Series (IMTS) forecasting. Unlike prior approaches that rely on resampling to generate coarse-grained sequences, ReIMTS preserves the original timestamps and recursively splits each sample into subsamples with progressively shorter time spans. An irregularity-aware fusion mechanism is introduced to integrate information across scales, enabling the model to capture both global and local temporal dependencies without destroying sampling patterns.

**Strengths:**

1. The paper is clearly written and logically structured, providing background on multi-scale modeling for irregular time series.

2. The proposed irregularity-aware fusion is intuitive and aligns well with the challenges of IMTS.

3. Experimental evaluation is comprehensive, covering multiple datasets and models, and demonstrates strong empirical results.

**Weaknesses:**

1. The novelty of the approach is somewhat limited in that the recursive multi-scale design can be viewed as a restructured version of existing patch-based or hierarchical multi-scale strategies.

2. The motivation for using a recursive structure, as opposed to other multi-scale fusion designs, is not clearly justified.

**Questions:**

1. Does the recursive decomposition introduce cumulative error across scales, and if so, how is this controlled or regularized?

2. How does ReIMTS handle dependencies among different variables in IMTS?

3. Could the authors clarify the specific motivation for adopting recursion instead of parallel or hierarchical multi-scale fusion?

4. Since the recursive splitting resembles patch-based segmentation, what distinguishes ReIMTS from existing hierarchical or patch-based multi-scale models?

---

> ### Author Response · Authors · 2025-11-24
>
> We thank the reviewer for recognizing the **clarity, intuition, and thorough evaluation** of our work as strengths!
> **We have updated our manuscript**, and your concerns are addressed as follows:
>
> ## W1 & Q4. Novelty discussions and comparisons with existing methods
>
> **A1**: Our contribution lies in achieving both:
>
> 1. **Preserving sampling patterns in irregular multivariate time series (IMTS)**: To our knowledge, ReIMTS is the *first* multi-scale learning method for IMTS that preserves sampling patterns. Prior approaches (Fig. 3 (b), 3 (d) of our revised manuscript) rely on resampling, which alters original timestamps, while patch-based methods (Fig. 3 (c)) often operate on a single time period.
> 2. **Applicable to most backbones**: Unlike prior multi-scale IMTS methods (Fig. 3 (d)) and patch-based methods (Fig. 3(c)), ReIMTS can be readily integrated with diverse backbones. It treats each backbone as an encoder–decoder pair without making assumptions about their internal implementations, enabling broad compatibility.
>
> Comparisons of our method with existing ones can be found in **A1.2 of our official comment to Reviewer MQ28 above**.
>
> ## W2 & Q3. Justification for the motivation to use the recursive structure
>
> We justify from two perspectives:
>
> ### Intuition perspective
>
> **A2.1**: Our recursive structure derives from the simple inspiration that **1 day = 24 hours = 24 × 60 minutes**.
> We think that time in the real world aligns better with the nested structure of recursion, instead of parallelism.
>
> ### Methodology perspective
>
> **A2.2**:
>
> **Compared with the parallel structure:**
>
> 1. **Recursive fusion naturally prioritizes closer observations over distant ones.**
>
>     In our framework, information from recent timestamps propagates through fewer fusion steps, while distant information must traverse multiple scales.
>     For example, in Fig. 2, variable $V_2$ at $t_9$ (scale 3) directly accesses $t_7$ (scale 2) via a single fusion step (scale 2→3), but reaches $t_1$ (scale 1) only through a longer chain (scale 1→2→3).
>     By contrast, the parallel design primarily rely on the final fusion stage.
>
> 2. **The recursive structure offers greater flexibility.**
>
>     - **Recursive structure:** The number of scale levels is controlled by recursion depth, making it easy to adjust when resizing the model.
>     - **Parallel structure:** Typically requires constructing all scale-specific modules during initialization [1], which limits flexibility when changing the number of scales.
>
> **Compared with the hierarchical structure that uses resampling:**
>
> These existing methods are depicted in Figure 3 (d) of our revised manuscript.
>
> 1. **Hierarchical resampling can disrupt sampling pattern information in IMTS.**
>
>     Resampling alters sampling timestamps of IMTS [2].
>     In contrast, our approach retains the original timestamps at every scale.
>
> ## Q1. Cumulative error discussions
>
> **A3**:
> We discuss its potential effects on the two main components of our method:
>
> 1. **Recursive splitting does not directly introduce cumulative error.**
>
>     Recursive splitting preserves each observation tuple $(t_i, z_i, v_i)$ exactly.
>     Thus, backbones at all scales operate on the *original observation values*, ensuring that no cumulative error is introduced in either the samples or their subsamples.
> 2. **Irregularity-aware fusion regulates information propagation through learnable scores and masks.**
>
>     The fusion process (Section 4.2) uses dataset-provided masks to prevent padding artifacts, while the learnable score $\alpha$ controls how much global information is fused with local representations.
>     Because fusion occurs in a high-dimensional feature space instead of sample space, defining “errors” is nontrivial without explicit ground truths.
>
> ## Q2. Learning variable dependency in ReIMTS
>
> **A4**: ReIMTS itself does not directly learn variable dependencies. Instead, it controls the time periods from which each backbone extracts dependencies.
> Backbone-specific details are provided in Appendix C of revised manuscript.
>
> Using ReIMTS+GraFITi as an example: GraFITi [3] models variable dependencies via a bipartite graph connecting variable nodes and timestamp nodes (see Fig. 2 (b) in the GraFITi paper).
> Dependencies are captured through two-step message passing: variable → shared timestamps → variable.
> In ReIMTS+GraFITi, variable-node embeddings are passed to the lower scale and fused, enabling multi-scale variable-dependency learning.
>
> [1] Woo, G., et al; "Unified Training of Universal Time Series Forecasting Transformers"; ICML 2024
>
> [2] Luo, Y., et al; "Hi-Patch: Hierarchical Patch GNN for Irregular Multivariate Time Series"; ICML 2025
>
> [3] Yalavarthi, V. K., et al; "GraFITi: Graphs for Forecasting Irregularly Sampled Time Series"; AAAI 2024
>
> ---
> **Thanks again for your careful review! Feel free to let us know if you have any further questions or concerns :-).**

---

> ### Author Response · Authors · 2025-11-27
> **Gentle Follow-up on Author Response**
>
> Dear Reviewer XY3z,
>
> We appreciate the time you have spent reviewing our submission.
> As the rebuttal phase is close to finishing, we would like to kindly ask whether the clarifications in our response resolve your main concerns, or if there is anything else we can help clarify.
>
> We fully understand that the review period is busy, and we are grateful for your efforts.

---

> ### Comment · Reviewer_XY3z · 2025-11-27
>
> Thank you for the authors’ efforts. The response has addressed most of my concerns, and the addition of Figure 3 provides a clearer illustration of the similarities and differences among existing methods. However, I still hold the view that the novelty of the proposed approach is limited to some extent. Since I have already provided a positive score, I will maintain my original rating.

---

### Official Review · Reviewer_MQ28 · 2025-11-01

**Soundness:** 2
**Presentation:** 3
**Contribution:** 3
**Rating:** 6
**Confidence:** 4

**Summary:**

The authors describe a new method to learn multi-scale representations for irregular time series data.

**Strengths:**

- original idea to learn multi-scale representations
- well written paper
- good results

**Weaknesses:**

- state-of-the-art comparison should be extended, especially in extending the discussions on strengths and weaknesses of previous work, clearly specifying the contributions and possible limitations of the new method proposed

**Questions:**

What are the limitations of your method, and for which dataset characteristics does it perform well, and for which not?

---

> ### Author Response · Authors · 2025-11-24
>
> We thank the reviewer for appreciating the **originality of our idea**, **the clarity of our writing**, and **the quality of our results** as strengths!
> **We have updated our manuscript**, and your concerns are addressed as follows:
>
> ## W1 & Q1. Pros and cons of our method VS existing ones
>
> **Figure 3** in revised manuscript illustrates method comparisons.
>
> ### Contributions of our new method
>
> **A1.1**: As illustrated in Figure 3 (a), our contribution lies in:
> 1. **Preserving sampling pattern**: As far as we know, our method ReIMTS is the *first* multi-scale learning approach for IMTS that can preserve informative sampling pattern information.
> 2. **Applicable to most backbones**: By viewing the backbone architecture as an encoder-decoder pair, our method does not restrict the actual implementation within the backbone's encoder or decoder.
>
> ### Comparisons with existing methods
>
> **A1.2**: We extend the discussions in **Section 4.4** of our revised manuscript, focusing on IMTS methods.
>
> **Patch-based methods for IMTS**
>
> They can be seen as variants of our method ReIMTS with only one scale level, as illustrated in **Figure 3 (c)**.
>
> **Pros:**
> 1. **Preserving the sampling pattern**
>
> **Cons:**
> 1. **May not be applicable to other backbones**: The special backbone designs in these methods can limit their ability to be migrated to other backbones. For example, PrimeNet [1] requires contrastive loss from its backbone during pre-training.
> 2. **Not directly scalable to multi-scale learning**: Most of them are limited to a single time period length during training.
>
> **Multi-scale methods via resampling in the sample space**
>
> This category of method is illustrated in **Figure 3 (b)**.
> Since IMTS methods in this category have been underexplored, we illustrate with Scaleformer [2] in our discussion.
>
> **Pros:**
> 1. **Applicable to many Transformer-based backbones**
>
> **Cons:**
> 1. **Disrupting sampling pattern of irregular time series**: In Figure 3 (b), $t_1$-$t_9$ at the top scale level are resampled as $t_1'$-$t_4'$ at the lower scale level.
>
> **Multi-scale methods via resampling in the representation space**
>
> As illustrated in Figure 3 (d), most existing multi-scale methods for IMTS fall within this category.
>
> **Pros**:
> 1. **Easy to understand during implementation**
>
> **Cons**:
> 1. **Disrupting sampling pattern of irregular time series**: As discussed above.
> 2. **Inflexible model architectures**: They require careful redesign of the backbone, hindering their broad applicability. For example, Warpformer redesign the self-attention layers with warping layers.
>
> ### Limitations of our method and possible solutions
>
> **A1.3**: We extend the discussion in Section 6:
> 1. **ODE-based models**
>
>     ODE-based models [3] may require additional theoretical explainability when used with our method.
> 2. **Diffusion-based models**
>
>     Some diffusion-based models generate noisy latent representations that may not be directly compatible with our method.
>     Possible solutions:
>     - Predict the original clean series during training
>
>         Predicting clean series has been evaluated on regular time series forecasting tasks [4], but not yet on irregular time series.
>
>     - Use ReIMTS as the denoising model during the diffusion process
>
>         For example, in CSDI [5], the denoising backbone learns observation representations, which can become multi-scale observation representations after applying ReIMTS.
>
> ## Q2. Discussion of dataset characteristics that favor our method
>
> **A2**:
> - **Conclusion:** ReIMTS excels on time series with longer maximum lengths, as demonstrated by the experiments on MIMIC‑IV and USHCN.
> - **Key insight:** Our recursive‑splitting approach turns long, irregular series into smaller windows, enabling the model to learn local dependencies even in highly sparse data.
> - **Evidence:**
>   - *Table 1* shows that MIMIC‑IV and USHCN have the longest maximum sequence lengths among the datasets.
>   - *Table 2* reports that ReIMTS yields the largest average performance gains across six backbones on these two datasets.
>
> [1] Chowdhury, R. R., et al; "PrimeNet: Pre-training for Irregular Multivariate Time Series"; AAAI 2023
>
> [2] Shabani, M. A., et al; "Scaleformer: Iterative Multi-scale Refining Transformers for Time Series Forecasting"; ICLR 2023
>
> [3] Rubanova, Y., et al; "Latent Ordinary Differential Equations for Irregularly-Sampled Time Series"; NIPS 2019
>
> [4] Yuan, X. and Qiao, Y.; "Diffusion-TS: Interpretable Diffusion for General Time Series Generation"; ICLR 2024
>
> [5] Tashiro, Y. et al; "CSDI: Conditional Score-based Diffusion Models for Probabilistic Time Series Imputation"; NeurIPS 2021
>
> ---
> **Thank you so much for your in-depth comments! Please let us know if you have any further questions:-).**

---

> ### Author Response · Authors · 2025-11-27
> **Gentle Follow-up on Author Response**
>
> Dear Reviewer MQ28,
>
> As the discussion period is approaching its conclusion, we would like to kindly check whether our response has addressed your main concerns. If any part of our clarification was unclear or requires additional detail, we would be happy to provide further explanation.
>
> We appreciate your time and consideration.

---

### Author Response · Authors · 2025-11-29
**Summary of Rebuttal Status and Score Changes**

Dear all,

We are sorry to hear about the recent data-leakage issue in OpenReview, and we fully support the corrective measures.

To help reduce the workload for the newly assigned AC, we provide below a concise summary of the rebuttal-stage status for our submission:

|Reviewer|Confidence (unchanged)|Initial Score $\rightarrow$ Score After Rebuttal|Timestamp of Reviewer Response (UTC+0)|
|---|---|---|---|
|MQ28|4|6 $\rightarrow$ **no reponse from reviewer**|-|
|XY3z|4|6 $\rightarrow$ 6(unchanged)|Nov. 27, 13:23|
|rfED|3|4 $\rightarrow$ **no reponse from reviewer**|-|
|4Jcx|4|6 $\rightarrow$ **8**|Nov. 27, 01:24|

We would also like to clarify that **we have never accessed or made use of any leaked data**. We learned about the OpenReview incident on Nov. 28, one day after the latest reviewer response.

Thanks for all the efforts in reassessment and for ensuring a fair review process.

Best regards,

Authors

---

### Public Comment · ~Boyuan_Li4 · 2026-03-13
**Correction to Experimental Results**

Dear all,

We recently identified and corrected a bug in our code affecting the USHCN dataset. The issue originated from an [earlier commit](https://github.com/Ladbaby/PyOmniTS/commit/ca9cd3c555451f8383751ebb72fdf7a446cc80b0) in our PyOmniTS code repository and has now been fixed in a [subsequent commit](https://github.com/Ladbaby/PyOmniTS/commit/d227e350397694482dbb1382f8b21a05ee1d8f4c). Following this correction, we reran all experiments on the USHCN dataset and updated the corresponding results in the camera-ready version of the paper.

**Importantly, the overall conclusions remain unchanged: our method, ReIMTS, consistently improves performance.**

Thanks for your understanding.

Best regards,

The Authors

---

### Meta-Review · Area_Chair_Jw7p · 2026-01-07

**Summary:**

This paper proposes a recursive multi-scale framework, named ReIMTS, for irregular multivariate time series forecasting that preserves original timestamps and fuses representations across scales in a backbone-agnostic manner. Specifically, it splits samples recursively and employs an irregularity-aware representation fusion mechanism.

Reviewers generally acknowledge that i) this paper is well-written; ii) achieves empirically strong results across multiple datasets and backbones; iii) original idea to learn multi-scale representations.

During the rebuttal, the authors have added a method taxonomy/figure comparing patch-based and resampling-based multi-scale approaches. Besides, they have provided additional downstream classification results to support the representation learning claim. They have also addressed presentation concerns with a notation overhaul and restructuring, and conducted fairness experiments (hyperparameters, backbone modifications). The limitations of the proposed method and possible solutions are also presented

The main remaining concern is technical novelty. Specifically, one reviewer maintains that the recursive multi-scale design may be viewed as a restructured version of existing patch-based or hierarchical multi-scale strategies. To address this concern, the authors argue that the key distinction is: i) preserving sampling patterns in irregular multivariate time series (IMTS) and ii) applicable to most backbones. Specifically, the proposed method treats each backbone as an encoder-decoder pair without making assumptions about its internal implementations.

In summary, considering the strong empirical validation and improved explanations, I lean towards acceptance and encourage the authors to refine the novelty framing and articulate limitations/when the approach is most beneficial.

**Reviewer Concerns:**

The Addressed Concerns:
+ Clearer positioning against prior work
+ Justification of the recursive design
+ Clarification of notation and concepts
+ Evidence for comparison fairness
+ Efficiency analysis
+ Additional validation beyond forecasting

However, two higher-level concerns remain:
+ Lingering skepticism about the degree of novelty
+ Residual uncertainty about practical robustness and generality, particularly regarding tuning dependence and resource usage across all backbones and settings

**Reviewer Scores:**

+ Reviewer MQ28 would like to keep his/her score.
+ Reviewer XY3z would like to keep his/her score.
+ Reviewer rfED would like to keep his/her score.
+ Reviewer 4Jcx would like to increase his/her score.

---

### Decision · Program_Chairs · 2026-01-26

Accept (Poster)